



# The MOPITT Version 9 CO Product: Sampling Enhancements and Validation

Merritt Deeter[1], Gene Francis[1], John Gille[1], Debbie Mao[1], Sara Martínez-Alonso[1], Helen Worden[1], Dan Ziskin[1], James Drummond[2,3], Róisín Commane[4], Glenn Diskin[5], and Kathryn McKain[6,7]

[1]Atmospheric Chemistry Observations and Modeling Laboratory, National Center for Atmospheric Research, Boulder, CO, USA
[2]Department of Physics, University of Toronto, Toronto, Ontario, Canada
[3]Department of Physics and Atmospheric Science, Dalhousie University, Halifax, Nova Scotia, Canada
[4]Department of Earth & Environmental Sciences, Lamont-Doherty Earth Observatory, Columbia University, Palisades, NY, USA
[5]Langley Research Center, NASA, Hampton, VA, USA
[6]Global Monitoring Laboratory, National Oceanic and Atmospheric Administration, Boulder, CO, USA
[7]Cooperative Institute for Research in Environmental Sciences, University of Colorado, Boulder, CO, USA

**Correspondence:** Merritt N. Deeter (mnd@ucar.edu)

**Abstract.**

Characteristics of the Version 9 (V9) MOPITT ("Measurements of Pollution in the Troposphere") satellite retrieval product for tropospheric carbon monoxide (CO) are described. The new V9 product includes many CO retrievals over land which, in previous MOPITT product versions, would have been discarded by the cloud detection algorithm. Globally, the number

of daytime MOPITT retrievals over land has increased by 30-40% relative to the Version 8 product, although the increase in retrieval coverage exhibits significant geographical variability. Areas benefiting from the improved cloud detection performance include (but are not limited to) source regions often characterized by high aerosol concentrations. The V9 MOPITT product also incorporates a modified calibration strategy for the MOPITT near-infrared (NIR) CO channels, resulting in greater temporal consistency for the NIR-only and thermal infrared-near infrared (TIR-NIR) retrieval variants. Validation results based on in-

situ CO profiles acquired from aircraft in a variety of contexts indicate that retrieval biases for V9 are typically within the range of ±5% and are generally comparable to results for the V8 product.

## 1   Introduction

MOPITT ("Measurements of Pollution in the Troposphere") is an instrument on the NASA Terra satellite which was launched Dec. 18, 1999. Measurements made by MOPITT's gas correlation radiometers (Drummond, 1989; Drummond et al., 2010)

operating in both thermal-infrared (TIR) and near-infrared (NIR) spectral bands enable retrievals of CO mixing ratio vertical profiles and total column values. The MOPITT instrument has produced a unique long-term data record well suited for a variety of applications. MOPITT CO products are used, for example, to forecast air quality (Inness et al., 2015), estimate CO emissions (Pechony et al., 2013; Zheng et al., 2018; Nechita-Banda, 2018; Gaubert et al., 2020), and validate other satellite products

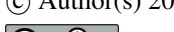



(Martínez-Alonso et al., 2014, 2020). Over the last two decades, MOPITT retrieval products have improved continuously as
knowledge has improved regarding the instrument, radiative transfer modeling, and geophysical variables (Worden et al., 2014;
Deeter et al., 2017).

MOPITT retrievals of CO volume mixing ratio (VMR) are generated with an optimal estimation-based retrieval algorithm
(Deeter et al., 2003). CO retrievals are based on a log(VMR) state vector (Deeter et al., 2007a) and are performed on a retrieval
grid with ten pressure levels (surface, 900 hPa, 800 hPa, ..., 100 hPa). Retrieval layers, used internally in the MOPITT retrieval
algorithm, are defined by the layers between each level in this grid and the next-highest level in the grid (Francis et al., 2017).
Thus, for example, the surface-level retrieval product actually represents the mean VMR for the layer between the surface and
900 hPa. Retrieved CO total column values are calculated directly from the CO profile and are not retrieved independently. A
priori CO profiles are derived from a model climatology based on the Community Atmosphere Model with Chemistry (CAM-
chem) chemical transport model (Lamarque et al., 2012), and vary seasonally and geographically; the a priori climatology
used for V9 products is identical to the climatology used for processing MOPITT Version 6, Version 7, and Version 8 products
(Deeter et al., 2014, 2017, 2019).

All MOPITT CO retrievals are based on a specific subset of the Average (A) and Difference (D) radiances from MOPITT
channels 5, 6, and 7; each channel is associated with a particular TIR or NIR gas correlation radiometer (Drummond et al.,
2010). TIR-only retrievals are based on the 5A, 5D, and 7D radiances in the 4.7 $\mu$m band, whereas NIR-only retrievals are based
solely on the ratio of the 6D and 6A radiances in the 2.3 $\mu$m band. MOPITT TIR-only retrievals are typically most sensitive
to CO in the mid- and upper-troposphere, except in scenes characterized by strong thermal contrast (Deeter et al., 2007b).
MOPITT NIR-only retrievals are most useful for retrievals of CO total column (Deeter et al., 2009; Worden et al., 2010).
Unique "multi-spectral" TIR-NIR retrievals exploit the 5A, 5D, 7D, 6D, and 6A radiances. This variant offers finer vertical
resolution than the TIR-only and NIR-only variants, and features the greatest sensitivity to CO in the lower troposphere (Deeter
et al., 2013). However, because NIR measurements rely on reflected solar radiation, the benefits of the TIR-NIR variant are
limited to daytime MOPITT observations over land.

This manuscript describes features of the new MOPITT V9 product which should be relevant to a wide spectrum of users.
Changes to the processing algorithms used to produce the V9 CO product are discussed in Section 2. These include significant
changes to (1) the method used to calibrate MOPITT's NIR radiances and (2) the cloud detection algorithm. Revisions to
the cloud detection algorithm resulting in significantly enhanced retrieval coverage were described and analyzed previously
in Deeter et al. (2021). V9 validation results based on in-situ measurements acquired from aircraft are compared with corre-
sponding V8 validation results in Section 3. Changes in retrieval sampling characteristics due to the revised cloud detection
algorithm and their impacts are analyzed in Section 4. Finally, conclusions are presented and discussed in Section 5.



## 2 Version 9 Algorithm Revisions

### 2.1 Calibration

Calibration of MOPITT's NIR radiances (6A and 6D) relies on a two-point calibration scheme involving both cold-calibration ("cold-cal") and hot-calibration ("hot-cal") events. Cold-cals are performed by pointing the scanning mirrors to space and occur many times per day. In contrast, hot-cals are typically performed annually as they require the execution of special instrument operations during which the internal blackbody is heated to $\sim$ 460 K (Drummond et al., 2010). Ideally, NIR-channel radiances are calibrated using hot-cals occurring both before and after the time of observation. While this method is feasible in retrospective processing mode (i.e., processing previous years of data), it is not possible in forward processing mode (i.e., when processing recently acquired observations). Thus, in forward processing mode, only information from the most recent hot-cal is used to calibrate MOPITT's NIR radiances. Comparisons of NIR-only retrieval products generated in retrospective and forward processing modes may exhibit significant differences (10% to 20%) in total column results, with the retrospectively processed data being more reliable (Deeter et al., 2017). Therefore, because of the degraded quality of MOPITT products processed in forward processing mode, V8 and V9 products generated in this manner are labeled as "beta" products to distinguish them from standard archival products. Beta products are eventually replaced by standard archival files following the next hot-cal. Typically, this occurs no more than a year after the time of a particular observation (depending on the date of the most recent hot-cal). Thus, beta products are considered provisional and should not be exploited for quantitative analyses.

For V9, the NIR calibration methodology for retrospective processing has been significantly revised. Hot-cals are typically performed annually, usually in March, in conjunction with a decontamination procedure; the entire series of instrument operations typically requires 12-13 days. In most years, hot-cals are executed both immediately before and after the decontamination procedure. For previous MOPITT products, including V8, NIR calibration for archival (non-beta) products relied on the closest bracketing hot-cals such that, usually, NIR radiances for a given date were calibrated using the most recent previous post-decontamination hot-cal and the next pre-decontamination hot-cal. For example, for V8, NIR radiances observed between March 5, 2016 and March 5, 2017 were calibrated using information from the post-decontamination hot-cal on March 4, 2016 and the pre-decontamination hot-cal on March 6, 2017.

However, it was recently discovered that this NIR calibration strategy often results in a growing retrieval bias in the NIR-only products over the period between the two hot-cals used for calibration. As illustrated in Fig. 1, this time-dependent bias is most obvious when comparing TIR-only and NIR-only CO products immediately before and after a particular hot-cal/decontamination cycle. Timeseries plots of daily-mean CO total column values are shown in the top panel for the V9 TIR-only (V9T), V8 NIR-only (V8N) and V9 NIR-only (V9N) products for all daytime retrievals over land regions between 60° S and 60° N. Timeseries are shown in the bottom panel for $\Delta$CO total column values obtained by subtracting daily-mean V9T CO total column values from corresponding V8N and V9N daily-mean values. Although NIR-only and TIR-only retrievals are characterized by different vertical sensitivities, and are therefore not expected to agree precisely, V9T total column values are a useful reference because they are unaffected by NIR calibration issues. Thus, TIR-only and NIR-only CO total column values averaged over large spatial scales should be expected to exhibit a very similar annual cycle.





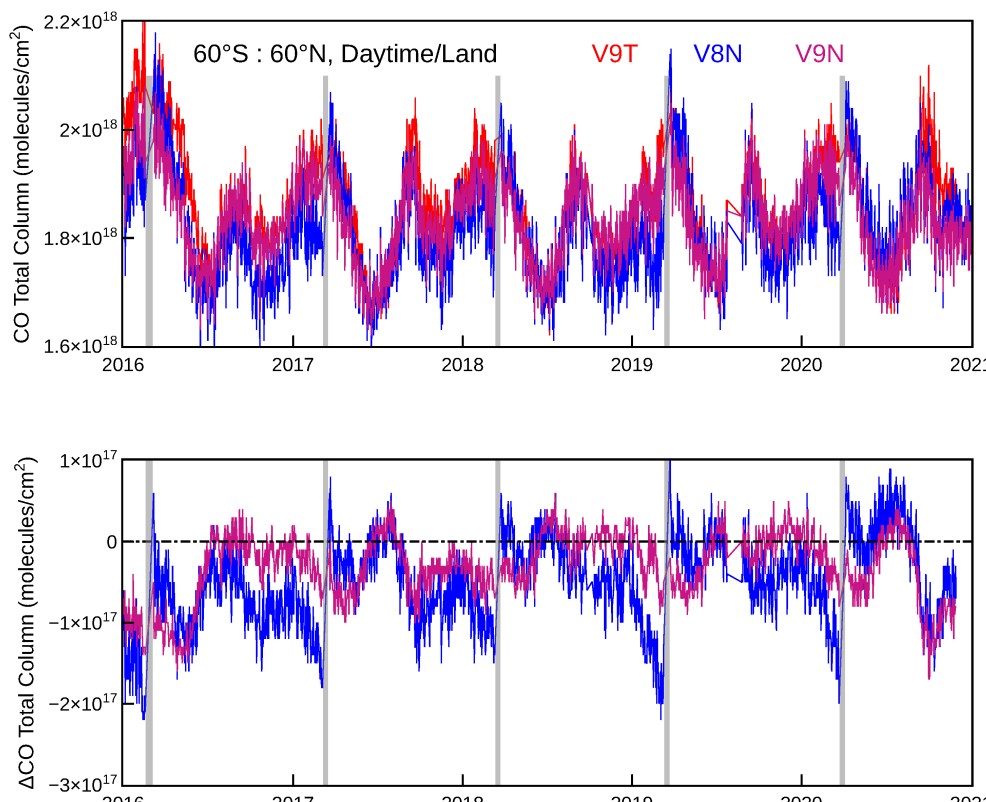

**Figure 1.** Timeseries comparisons of mean CO total column and $\Delta$CO total column for daytime/land retrievals between $60^\circ$S and $60^\circ$N (as described in Sec. 2.1) for the V9T, V8N and V9N variants. $\Delta$CO total column timeseries in lower panel are obtained by subtracting the V9T total column timeseries (plotted in red in the top panel) from the V8N (blue) and V9N (purple) timeseries. Vertical gray bars indicate periods during which the annual hot-calibration and decontamination procedures were performed. Discontinuities in $\Delta$CO total column for dates just before and after the hot-cal/decontamination events for the V8N variant (blue) are largely resolved for the V9N variant (purple).

Vertical gray bars shown in the upper and lower panels of Fig. 1 indicate periods during which the annual hot-calibration and decontamination procedures were performed. For each of the years shown from 2016 to 2020, the $\Delta$CO total column timeseries for V8N (plotted in blue) exhibits a physically unrealistic discontinuity when comparing dates just before a pre-decontamination hot-cal with dates just after the post-decontamination hot-cal several weeks later. For example, in 2019, $\Delta$CO total column for V8N increased from about $-2\times10^{17}$ molecules/cm$^2$ just before the pre-decontamination hot-cal to close to 0 just after the post-decontamination hot-cal. While the physical source of this discontinuity is not yet fully understood, it





suggests that the pre- and post-decontamination hot-cals are not consistent with each other and are not equally useful for

calibration.

Experiments were performed to develop an improved NIR calibration strategy for V9. It was found that the typical discontinuity in $\Delta$CO total column values before and after the hot-cal and decontamination cycle was greatly reduced when only post-decontamination hot-cals were used for calibration. The $\Delta$CO total column timeseries using this strategy, which was implemented for V9N operational processing, is plotted in purple in the bottom panel of Fig. 1. For each of the years shown,

the improved stability of the V9N product compared to V8N is clearly evident. Additional details regarding the specific hot-cals used for NIR calibration in V9 over the entire MOPITT mission will be reported in a forthcoming revision of the L0-L1 Algorithm Theoretical Basis Document (ATBD).

## 2.2 Radiative Transfer Modeling

The operational MOPITT radiative transfer model, known as MOPFAS, is updated monthly with information describing the

mean instrument state for that month, including the pressures and temperatures in the gas correlation cells (Edwards et al., 1999; Deeter et al., 2013). For V9, operational modeling of the MOPITT Pressure Modulation Cell (PMC) radiances (7A and 7D) now also includes monthly updated values for optical depth. The operational radiative transfer model for V9 is based on HITRAN12 (Rothman et al., 2013), which is the same version of HITRAN used for MOPITT V7 and V8 processing.

The MOPITT retrieval algorithm exploits radiance bias correction factors to compensate for relative biases between simu-

lated radiances calculated by MOPFAS and actual calibrated Level 1 radiances from the instrument. Radiance bias correction factors compensate for a variety of potential bias sources including errors in instrumental specifications, forward model errors related to the development of MOPFAS, errors in assumed spectroscopic data, and geophysical errors (Deeter et al., 2014). Within the retrieval algorithm, these correction factors are applied by scaling the simulated radiances produced by MOPFAS each time it is executed.

As introduced in V8 processing, a radiance bias correction is based on a parameterization involving both (1) the date of the MOPITT observation and (2) the water vapor total column at the time and geographic location of the MOPITT observation, as derived from the MERRA-2 (https://gmao.gsfc.nasa.gov/reanalysis/MERRA-2/) water vapor profiles needed to execute MOPFAS (Deeter et al., 2019). Within the retrieval software, the radiance bias correction factors for V8 and V9 are calculated using the relation

$$R^i = R_0^i + R_t^i N_{dys} + R_w^i WV \tag{1}$$

where $R^i$ is the multiplicative radiance correction factor to be applied to the model-simulated value for radiance $i$, $N_{dys}$ is the number of elapsed days since January 1, 2000, $WV$ is the water vapor total column (or "precipitable water vapor", expressed in molecules/cm$^2$) determined from the MERRA-2 reanalysis (temporally and spatially interpolated to the time and location of the MOPITT observation), and $R_0$, $R_t$, and $R_w$ are the empirically-determined parameters which effectively minimize overall retrieval bias, bias drift and bias water vapor sensitivity.





Values of $R_0$, $R_t$, and $R_w$ for the 5A, 5D, 6D, and 7D radiances used for V8 and V9 operational processing are listed in Table 1. (Since the use of MOPITT's NIR radiances in the retrieval algorithm only involves the ratio of the 6D and 6A radiances, values of $R_0$, $R_t$, and $R_w$ for the 6A radiance are not optimized as they are for the other radiances. Thus, for 6A, $R_0$ is set to 1, while $R_t$ and $R_w$ are both set to 0.) V9 values are identical to the corresponding values used for V8 processing, except for the $R_0$ and $R_t$ values for 6D and 7D. V9 values of $R_0$ and $R_t$ values were re-optimized for 6D because of the revised

calibration scheme described in Section 2.1. Values of $R_0$ and $R_t$ values were re-optimized for 7D due to the forward model corrections related to PMC modeling. The methods used to optimize the $R_0$ and $R_t$ values for 6D and 7D are described in Deeter et al. (2019). As indicated in Table 1, V9 radiance bias correction factors for 7D are smaller than the corresponding correction factors for V8, suggesting that the PMC model revisions in MOPFAS implemented for V9 resolved a substantial component of the discrepancy between observed and model-calculated radiances for Channel 7.

**2.3   Cloud Detection**

Because the MOPITT radiative transfer model simulates radiances only in clear-sky conditions, MOPITT observations affected by clouds are not used in Level 2 retrieval processing. The clear/cloudy determination is performed by a cloud detection algorithm which involves both MOPITT's thermal-channel radiances and information from the MODIS ("Moderate Resolution Imaging Spectroradiometer") cloud mask product (Warner et al., 2001; Francis et al., 2017). With respect to the MOPITT

thermal-channel test, the ratio of the observed MOPITT Channel 7 Average radiance and the corresponding model-calculated value is compared to a predefined global threshold value. If the radiance ratio is less than the threshold value, that MOPITT observation is flagged as cloudy. For V9, the radiance ratio for each MOPITT retrieval is reported in the new diagnostic "MOPCld Rad Ratio".

The overall outcome of the MOPITT cloud detection algorithm for a particular retrieval is described by the "Cloud Descrip-

tion" diagnostic in the Level 2 files. The Cloud Description diagnostic values (1-6) are defined as follows.

   1: MOPITT clear, MODIS cloud mask unavailable

   2: MOPITT clear, MODIS cloud mask clear

   3: MOPITT cloudy, MODIS cloud mask clear

   4: MOPITT clear, MODIS cloud mask indicates low clouds only

145       5: Polar regions, MODIS cloud mask clear (no MOPITT test)

   6: MOPITT clear, MODIS cloudy

This last class (6) was first introduced in the V7 product and was applied only to ocean scenes as a response to declining quality in the MODIS cloud mask (Deeter et al., 2017).

For the V9 product, two significant changes were implemented in the revised cloud detection algorithm (Deeter et al., 2021).

The first change is related to the interpretation of the MODIS cloud mask, whereas the second change concerns the treatment of observations deemed cloudy by the MODIS cloud mask but clear by the MOPITT thermal channel test. Together, these changes significantly increase MOPITT retrieval coverage over land.





The MODIS cloud mask reports one of four possible outcomes for each MODIS 1-km pixel: Cloudy, Uncertain, Probably Clear, or Clear. An individual MOPITT pixel typically encloses $\sim 500$ MODIS 1-km pixels. Prior to V9, the MOPITT cloud

detection algorithm interpreted the Probably Clear and Clear outcomes as clear and treated the Cloudy and Uncertain outcomes as cloudy. If at least 95% of the MODIS cloud mask pixels enclosed within a given MOPITT pixel indicated either Probably Clear or Clear, that MOPITT pixel was considered clear according to MODIS. For V9 processing, the MODIS cloud mask test was relaxed to treat Uncertain MODIS pixels as clear in the same manner as Clear and Probably Clear MODIS pixels. This change was motivated by the observation that such MODIS pixels can often be found in apparently cloudless but heavily

polluted scenes (Deeter et al., 2021).

For V8 and earlier MOPITT products, observations over land were typically discarded if the MODIS cloud mask indicated clouds. In V9, however, observations over land are only discarded if both the MODIS cloud mask and MOPITT radiance tests indicate the presence of clouds; this change was introduced earlier for observations over the ocean, beginning with V7 products. It allows MOPITT retrievals in cases where the MODIS cloud mask tests indicate clouds (or are ambiguous) while

the MOPITT TIR radiances are consistent with clear-sky conditions. Consequently, this change should allow the retrieval of scenes for which clouds in the MOPITT field of view have a negligible effect on the MOPITT radiances. MOPITT retrievals for which the MODIS cloud mask considers the observation to be cloudy while the MOPITT thermal channel test passes the observation as clear are assigned the Cloud Description index of 6 and can therefore be analyzed separately from retrievals where MODIS determined the scene to be clear. Prior to V9, this value for the Cloud Description index was only allowed for

observations over the ocean.

Finally, a minor change was also made in the revised cloud detection algorithm regarding cloud index 4 (MOPITT clear, MODIS indicating low clouds). In the revised algorithm, this index is only applied to observations over the ocean, where low clouds are more reliably detected. Retrievals over land which would have been assigned a cloud index value of 4 in the V8 algorithm are assigned a cloud index value of 6 in V9. Thresholds for the MODIS cloud mask and MOPITT thermal channel

tests for V9 are unchanged relative to the values used for the MOPITT Version 8 product, i.e., the MODIS clear-sky fraction threshold is set to 0.95 and the MOPITT radiance ratio threshold is set to 1.00.

In addition to the "Cloud Description" diagnostic, a separate diagnostic is provided for each retrieval in the Level 2 product file to quantify the results of the various cloud tests applied to the set of MODIS Cloud Mask pixels matched to each MOPITT pixel (Francis et al., 2017). This diagnostic, which has been revised for V9, may be of use for analyzing potential retrieval

biases associated with particular types of scenes. For V9, elements of the 12-element "MODIS Cloud Diagnostics" floating point vector indicate:

(1) Number of valid MODIS pixels

(2) Percentage of cloudy MODIS pixels

(3) Percentage of clear MODIS pixels, test 1

(4) Percentage of clear MODIS pixels, test 2

(5) Percentage of clear MODIS pixels, test 3

(6) Average value of "sun glint" MODIS flag

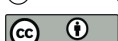



(7) Average value of "snow/ice background" MODIS flag

(8) Average value of "non-cloud obstruction" MODIS flag

(9) Average value of "IR threshold test" MODIS flag

(10) Average value of "IR temperature difference tests" MODIS flag

(11) Average value of "visible reflectance test' MODIS flag

(12) Fraction of valid MODIS pixels

Elements 3, 4, and 5 of the Cloud Diagnostics vector report the percentage of clear-sky MODIS pixels within the MOPITT pixel according to three tests with varying levels of confidence. Test 1 (vector element 3) reports the clear-sky percentage based solely on Clear outcomes for the MODIS cloud mask and is therefore the strictest test. Test 2 (element 4) reports the clear-sky percentage considering both Clear and Probably Clear MODIS cloud mask outcomes as clear, and corresponds to the cloud mask test used in prior versions of the MOPITT cloud detection algorithm. Finally, test 3 (element 5) reports the clear-sky percentage considering Clear, Probably Clear, and Uncertain MODIS cloud mask outcomes as clear. For V9, this last test actually determines whether MODIS classifies the MOPITT pixel as clear or cloudy. Elements 3 and 5 in the Cloud Diagnostics vector represent information not previously included in the MOPITT product.

## 3  Validation

Retrieval validation results for the V9 product are compared with corresponding results for the V8 product below. Validation results are based on quantitative comparisons of MOPITT retrieval products (CO VMR profiles and total columns) with in-situ vertical profiles measured from aircraft. In-situ measurements are assumed to be exact and representative of a defined region surrounding the sampling location. When making quantitative comparisons of MOPITT retrieved CO profiles and in-situ profiles, the in-situ data must be transformed to account for the effects of smoothing error and inclusion of a priori information (Deeter et al., 2003). Neglecting retrieval error, the relationship between the true profile $x_{true}$, a priori profile $x_a$ and retrieved profile $x_{rtv}$ is expressed by the equation

$$x_{rtv} = x_a + A(x_{true} - x_a) \tag{2}$$

where $A$ is the retrieval averaging kernel matrix (Rodgers, 2000). For consistency with the MOPITT retrieval algorithm, the vector quantities $x_{true}$, $x_a$ and $x_{rtv}$ are expressed in terms of log(VMR) rather than VMR.

Previously reported validation results based on a set of aircraft profiles over the Amazon Basin demonstrated that retrieval biases for the V8 TIR-only product and an experimental product incorporating the cloud detection revisions described in Section 2.3 were within about 3% at all levels (Deeter et al., 2021). However, since the disparities were similar to the estimated accuracy of the in-situ measurements, the difference in biases was not considered significant. Below, we compare V8 and V9 validation results over a much larger set of aircraft profiles drawn from both a long-term measurement program operated by NOAA and several field campaigns.



## 3.1 NOAA Aircraft Profiles

V8 and V9 validation results reported below are based on a large set of CO vertical profiles measured by the NOAA Global
Monitoring Laboratory using an airborne flask-sampling system followed by laboratory analysis (Sweeney et al., 2015). Typical
in-situ profiles are derived from a set of twelve flasks acquired as the aircraft descends. Reproducibility of the laboratory-
measured CO dry-air mole fractions, which are measured by either a vacuum UV–resonance fluorescence spectrometer or
a reduction gas analyzer, is better than 1 ppb. Total uncertainty values for the flask measurements increase monotonically
with CO mole fraction from $\sim 1.2$ ppb at 100 ppb to $\sim 3.5$ ppb at 500 ppb (https://gml.noaa.gov/ccl/ccl_uncertainties.html).
All NOAA flask sample profiles were calibrated using the WMO CO X2014A scale (https://gml.noaa.gov/ccl/co_scale.html).
Results reported below are based on NOAA vertical profiles obtained from flights at 21 fixed sites (mainly over North America)
between 2000 and 2020. The consistency, long record and high accuracy characterizing this set of profiles is the basis for its use
in optimizing the radiance bias correction factors and for quantifying long-term changes in MOPITT retrieval biases (Deeter
et al., 2003, 2019).

For matching MOPITT retrieved profiles with the NOAA in-situ profiles, a maximum separation of 50 km was employed
and a maximum of 12 hours was allowed between the time of the MOPITT observation and sampling time of the in-situ data.
In order to obtain a complete validation profile for comparison with MOPITT retrievals, each in-situ profile was extended
vertically above the highest-altitude in-situ measurement using the CAM-chem chemical transport model (Lamarque et al.,
2012) and then resampled to the standard pressure grid used for the MOPITT operational radiative transfer model (Martínez-
Alonso et al., 2014). Validation results for the MOPITT 100 hPa retrieval level are not reported below, since in-situ data are
generally unavailable from aircraft for the atmospheric layer above this height.

Validation results derived from the NOAA aircraft flask samples for the V8 and V9 TIR-only, NIR-only and TIR-NIR
variants are compared in Fig. 2. Validation statistics for total column and alternating retrieval levels (surface, 800 hPa, 600
hPa, ...) are also summarized in Table 2. The left panel in Fig. 2 shows the mean retrieval bias versus pressure level and is
obtained by calculating the mean log(VMR) retrieval error over all MOPITT retrievals matched to one of the NOAA in-situ
profiles according to the matching criteria described above. Retrieval error is calculated for each retrieval by subtracting the
simulated in-situ based value (as calculated using Eq. 2) from the actual retrieved value. Retrieval bias values are converted
from $\Delta(\log(\text{VMR}))$ to % as described in Deeter et al. (2017). The panel on the right side of Fig. 2 presents the retrieval bias
drift at each pressure level as calculated using a least-squares fit to log(VMR) retrieval error as a function of time.

Overall retrieval bias values for the V9 TIR-only variant based on the NOAA profile set are generally in the range of a few
% and are comparable to corresponding V8 TIR-only values. The mean total column bias for V9, listed in Table 2, is slightly
smaller than for V8 ($9.69 \times 10^{15}$ molecules/cm$^2$ vs. $1.33 \times 10^{16}$ molecules/cm$^2$). Retrieval bias drift for the V9 TIR-only variant
is less than 0.2%/yr at all levels and is similar in magnitude to values for the V8 TIR-only variant. However, total column bias
drift is somewhat larger for V9 than for V8 ($1.52 \times 10^{15}$ molecules/cm$^2$/yr vs. $1.17 \times 10^{15}$ molecules/cm$^2$/yr).

As shown in Fig. 2, NOAA validation results for the V9 NIR-only variant are slightly worse than for the V8 NIR-only variant.
Nevertheless, for the V9 NIR-only variant, retrieval bias is still less than 1% at all levels and retrieval bias drift is generally





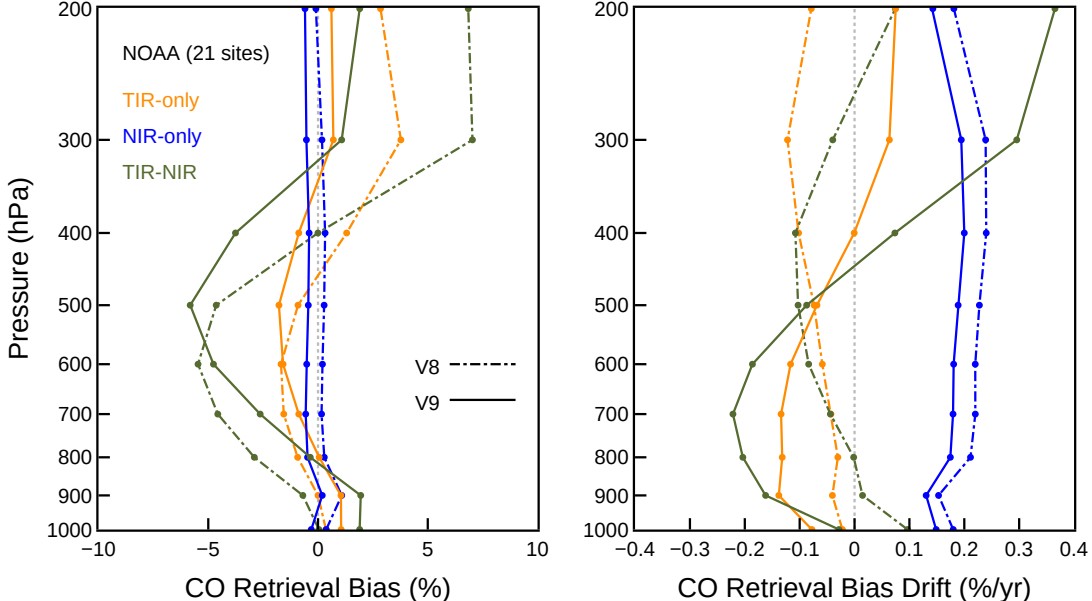

**Figure 2.** Comparison of V8 and V9 validation results based on the NOAA aircraft profile dataset.

less than 0.2%/yr at all levels. Total column bias and bias drift for the V9 NIR-only variant are $4.60 \times 10^{15}$ molecules/cm$^2$ and $3.27 \times 10^{15}$ molecules/cm$^2$/yr, both of which are improved relative to the V8 NIR-only variant.

Retrieval biases for the V9 TIR-NIR variant are generally larger (in magnitude) than for the V9 TIR-only and NIR-only variants, but are similar to values for the V8 TIR-NIR variant. Retrieval bias for the V9 TIR-NIR variant varies from -5.82% at 500 hPa to 1.90% at the surface. Total column bias is somewhat smaller for the V9 TIR-NIR variant compared to the V8 TIR-NIR variant ($1.60 \times 10^{16}$ molecules/cm$^2$ vs. $1.82 \times 10^{16}$ molecules/cm$^2$). Bias drift for the V9 TIR-NIR variant varies from -0.22%/yr at 700 hPa to 0.37%/yr at 200 hPa. V9 bias drift is smaller (in magnitude) than for the V8 TIR-NIR at the surface, but is larger than V8 bias drift values in both the lower troposphere (600-900 hPa) and upper troposphere (200-300 hPa). Total

column bias drift for V9 is also larger than for V8 ($-3.16 \times 10^{14}$ molecules/cm$^2$/yr vs. $-2.27 \times 10^{14}$ molecules/cm$^2$/yr), but is smaller than total column bias drift values for both the V9 TIR-only and V9 NIR-only variants.

     Standard deviation values are also listed in Table 2. Although this metric is often used to characterize random retrieval error, it is also influenced by limitations of the reference dataset used for validation. For example, the use of a single set of 12 flask measurements at discrete altitudes to fully represent the CO distribution sampled by MOPITT likely exaggerates the actual

retrieval error for several reasons including (1) fine-scale CO vertical variability not represented by the relatively coarse set of in-situ measurements, (2) horizontal CO variability within the co-location radius, (3) temporal CO variability during the delay between the in-situ sampling and MOPITT overpass and (4) the lack of in-situ measurements at high altitudes (e.g., above 10





km). Thus, the standard deviation values listed in Table 2 should be interpreted only as an upper bound for the actual random retrieval error. Alternative methods for analyzing random retrieval error will be the topic of a future study.

## 3.2 Cloud Index

As described in Section 2.3, a cloud index diagnostic (1-6) is included in the MOPITT Level 2 data files for each retrieved profile and indicates the manner in which that observation passed the cloud detection algorithm. V8 and V9 retrieval biases for each of the six cloud index subsets are analyzed in Appendix A. The analysis is based on the same NOAA profile set for which the aggregate validation statistics are shown in Fig. 2. Except for cloud index 1, which represents only $\sim 1\%$ of the analyzed data, results presented in the Appendix show that biases for the cloud index subsets are in the range of $\pm 5\%$ for the V9 TIR-only results, $\pm 2\%$ for the V9 NIR-only results, and $\pm 10\%$ for the V9 TIR-NIR results. Compared to the biases for the non-subsetted NOAA validation results (shown in Fig. 2), bias differences associated with the different cloud index values are generally no more than 2-3%.

## 3.3 Field Campaigns

The new V9 product was also separately validated using CO in-situ profiles measured during the HIPPO ("HIAPER Pole-to-Pole Observations"), ATom ("The Atmospheric Tomography Mission", https://espo.nasa.gov/atom) and KORUS-AQ ('The Korea-United States Air Quality Study", https://espo.nasa.gov/korus-aq) field campaigns. Both the HIPPO and ATom programs produced large sets of CO in-situ profiles over both the Northern and Southern Hemispheres, primarily over open ocean. Flights for HIPPO were conducted in five phases in 2009, 2010 and 2011 (Wofsy et al., 2011). ATom took place in four phases in 2016, 2017 and 2018. The KORUS-AQ campaign was conducted over the Korean peninsula (and vicinity) from April to June, 2016.

CO measurements used for validation for both HIPPO and ATom (https://doi.org/10.3334/ORNLDAAC/1747) were performed with the QCLS ("Quantum Cascade Laser Spectrometer") instrument (Santoni et al., 2014). CO measurements for KORUS-AQ were performed with the DACOM ("Differential Absorption Carbon monOxide Measurement") instrument (Sachse et al., 1987). In-flight calibration for both the QCLS and DACOM instruments involves the use of compressed gas cylinders from NOAA's Global Monitoring Laboratory with known CO concentrations. For ATom and KORUS-AQ, the calibration of these reference cylinders from NOAA was based on the WMO CO X2014A scale, whereas for HIPPO the calibration was based on the prior X2004 scale. For the HIPPO, ATom and KORUS-AQ CO measurements used below, potential drift in the reference cylinder CO mole fractions (https://gml.noaa.gov/ccl/co_scale.html) was addressed by calibrating the reference cylinders at NOAA's Central Calibration Laboratory both before and after the field campaign, and applying linear interpolation. For CO, the estimated precision of the QCLS instrument is 0.2 ppb (Santoni et al., 2014). For DACOM, the estimated precision is 1 ppb + 1% of the measured CO mole fraction (Sachse et al., 1987). A comparison of CO measurements obtained by QCLS and NOAA flasks during HIPPO indicated a negative bias of 2 ppb for QCLS (Santoni et al., 2014).

For matching MOPITT retrieved profiles with in-situ profiles, a maximum collocation radius of 50 km was employed for the KORUS-AQ profiles (like the NOAA profiles) whereas a value of 200 km was used for the HIPPO and ATom profiles. The larger radius for HIPPO and ATom was chosen since expected horizontal CO gradients are generally much smaller over the





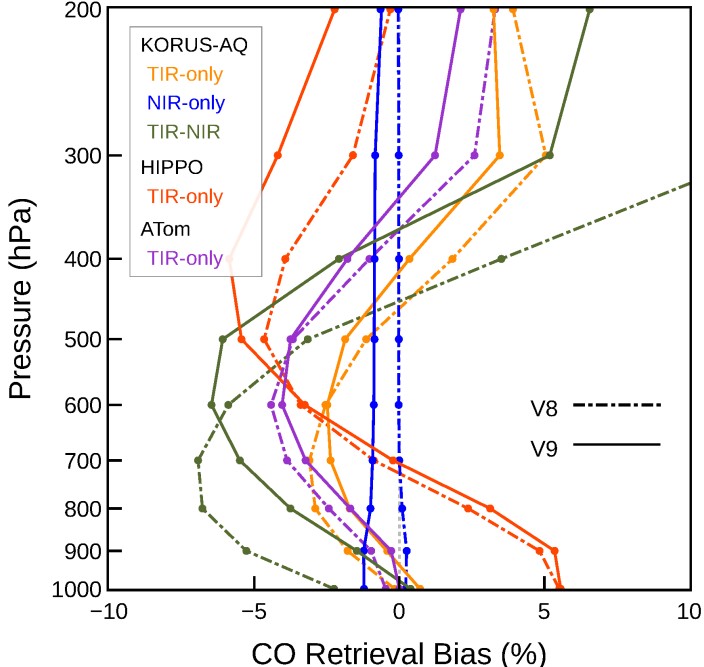

**Figure 3.** Comparison of V8 and V9 validation results based on CO profiles measured during the HIPPO, ATom and KORUS-AQ field campaigns.

open ocean than over continental regions. The influence of collocation criteria on MOPITT validation statistics was studied in Tang et al. (2020).

V8 and V9 TIR-only validation results for HIPPO and ATom are compared in Fig. 3 and Tables 3 and 4. V9 retrieval biases for HIPPO vary over the range of ±6%, while V9 retrieval biases for ATom vary from about -4% to 2%. With respect to total

column, biases for the V9 TIR-only product for the NOAA, HIPPO, and ATom (listed in Tables 2, 3 and 4) are $9.69 \times 10^{15}$, $-2.06 \times 10^{15}$, and $-1.22 \times 10^{16}$ molecules/cm$^2$, respectively. For both HIPPO and ATom, the range of observed biases (over the vertical profile) is larger than for the NOAA TIR-only profiles. To some degree, the smaller biases for the NOAA profiles is likely a consequence of using those profiles to obtain optimal radiance bias correction factors, as described in Deeter et al. (2019). Differences in biases for the NOAA, HIPPO, ATom and KORUS-AQ datasets could reflect either some type of

geographically variable retrieval bias in the MOPITT retrievals or differences in the characteristics of the in-situ measurements acquired during the field campaigns.

V8 and V9 validation results for KORUS-AQ are compared in Fig. 3 and Table 5. Differences between V8 and V9 retrieval biases for KORUS-AQ are generally similar to differences observed for the NOAA profile set. For example, in comparison to





V8, V9 TIR-only biases in the lower troposphere are shifted to slightly greater values in the lower troposphere and shifted to

slightly smaller values in the upper troposphere. The range of bias values over the CO profile for V8 and V9 is also similar. Biases for the V8 and V9 TIR-only, NIR-only and TIR-NIR variants for KORUS-AQ fall in the ranges ±4%, ±2% and ±7% respectively. Total column biases for the V9 TIR-only, NIR-only and TIR-NIR variants listed in Table 5 are somewhat larger than for the corresponding V8 variants (in contrast to the NOAA validation results).

## 4    Sampling Characteristics

Case studies presented in Deeter et al. (2021) illustrated the increased retrieval yield in selected scenes resulting from the cloud detection revisions described in Section 2.3. This previous analysis focused on the performance of the revised cloud detection algorithm in heavily polluted regions. Retrievals added because of the cloud detection revisions were found to be physically consistent with the retrieved CO in the rest of the scene. Below, we analyze the improved retrieval coverage in V9 products at global and regional spatial scales.

### 4.1    Zonal Means

Zonal totals of the numbers of daytime retrievals over land obtained for the V8 and V9 TIR-only variants for the month of July, 2017 are presented in the left panel of Fig. 4. Each plotted point indicates the total monthly number of daytime retrievals in a $10°$-wide latitude band. The plot illustrates a sharp increase in the number of daytime retrievals over land for V9, especially over the Northern Hemisphere. Globally, the total number of daytime retrievals over land increased by 41% from $9.84 \times 10^5$

for V8 to $1.36 \times 10^6$ for V9. Monthly totals of numbers of retrievals for V9 for other months which have been analyzed are typically 30-40% larger than for V8.

The panel on the right side of Fig. 4 compares V8 and V9 zonal-mean total column values for the same subsets of daytime retrievals over land analyzed in the left panel. The plot shows that the large relative increase in the number of daytime retrievals over land for V9 has a nearly negligible effect on the monthly-average total column zonal means. This finding suggests that

the retrievals added in V9 by virtue of the cloud detection algorithm changes described in Section 2.3 may not strongly affect large-scale features in the MOPITT product.

### 4.2    Sampling Frequency

The utility of MOPITT data for specific applications often depends on the temporal interval between observations. As illustrated below, a useful metric for this variable is retrieval sampling frequency (Deeter et al., 2021). We define retrieval sampling fre-

quency as the reciprocal of the mean sampling period, which is itself defined as the average number of days between retrievals acquired within a one-degree latitude by one-degree longitude grid cell, calculated over a specified period of observations. Thus, for a particular grid cell,

$$\nu_s = \tau_s^{-1} = (L_{obs}/N_{obs})^{-1} \qquad\qquad (3)$$





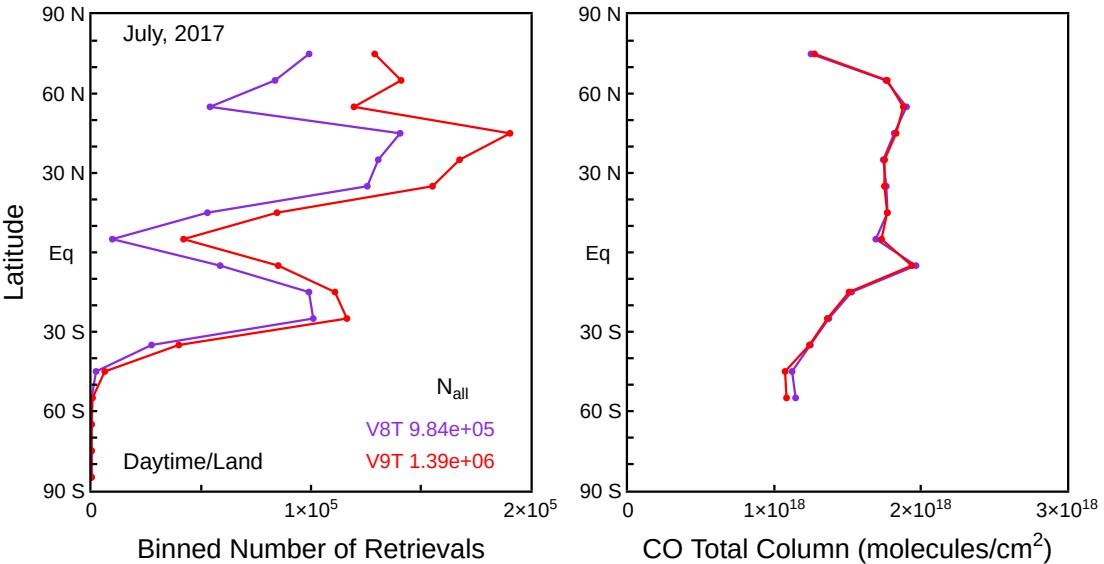

**Figure 4.** Comparison of zonal-mean values for number of retrievals (left) and mean CO total column (right) for V8 and V9 TIR-only variants based on daytime/land retrievals for July, 2017.

where $\nu_s$ is the retrieval sampling frequency, $\tau_s$ is the mean sampling period, $L_{obs}$ is the total length of the observation period (in days) and $N_{obs}$ is the number of days within that period which contain at least one MOPITT retrieval. In order to sample all

longitudes equally, sampling frequency should be calculated over periods of observations equal to integral multiples of Terra's 16-day orbital repeat cycle.

     Maps of daytime retrieval sampling frequency for V8 and V9 retrievals for South America are compared in Fig. 5. Retrieval sampling frequency was calculated for the period between September 1 and October 2, 2017, spanning two complete Terra orbital repeat cycles. No filtering was applied with respect to cloud index or any other parameter. Sampling frequency over

oceanic grid cells, which is not significantly different for the two cloud detection algorithms, is not shown. Grid cells for which the sampling frequency is exactly 0 (meaning that no retrievals were acquired over the entire 32-day observation period) are indicated by a cross covering the cell.

     As shown in Deeter et al. (2021), increased sampling frequency for V9 results from both of the cloud detection algorithm revisions described in Section 2.3. For V8 results shown in the left panel, sampling frequency varies widely from zero in

much of the extreme northern, easternmost and southwestern regions of South America to $\sim 0.3$ per day in parts of eastern South America and an area of western South America between $30°$ S and $20°$ S. For the V9 product, shown in the right panel, improved retrieval sampling frequency is indicated over most of the continent, but is most obvious in the regions where the





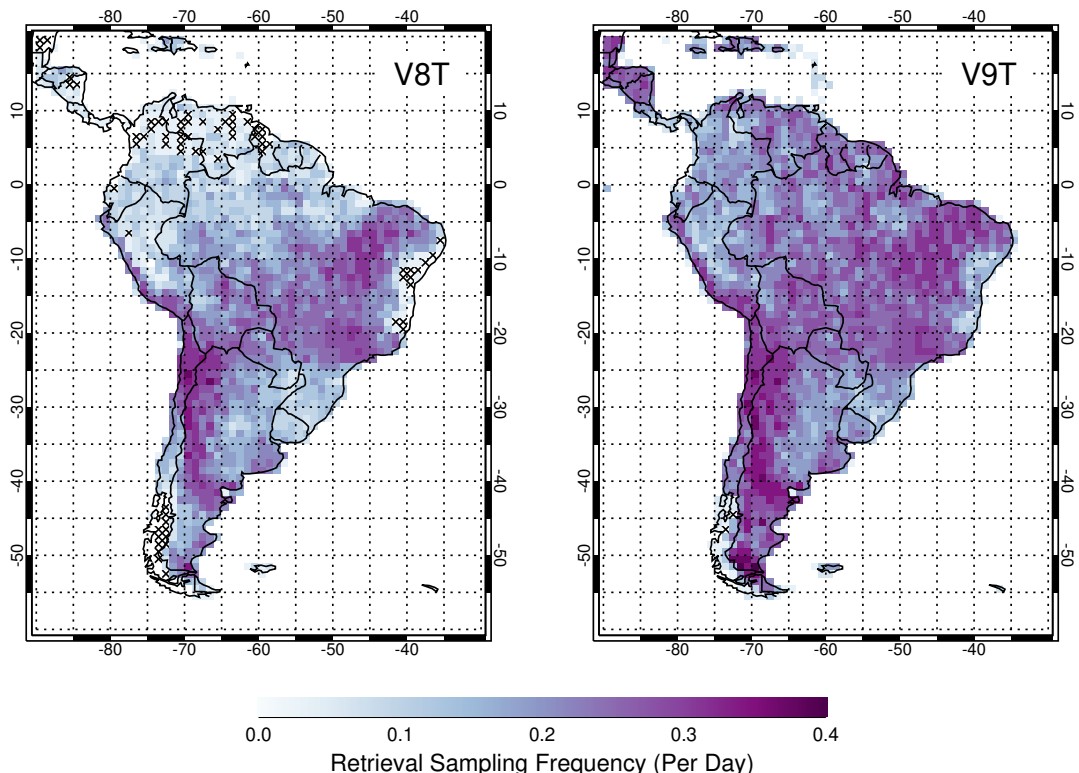

**Figure 5.** Comparison of maps of sampling frequency (defined in Section 4.2) for South America for V8 and V9 TIR-only variants based on daytime/land retrievals for September, 2017. No filtering was applied with respect to cloud index or any other parameter. Grid cells with enclosed crosses indicate a sampling frequency of exactly 0, meaning that no retrievals were obtained during the specified period.

V8 sampling frequency is the poorest, e.g., regions north of 5° S. Over this region, the mean sampling frequency increases by 127%, from 0.088 to 0.20 per day. Over the entire continent, the number of grid cells for which the retrieval sampling frequency

is exactly zero decreases sharply from 62 to 2.

Substantial improvements in sampling frequency for V9 are also observed for North America and Asia. V8 and V9 sampling frequency maps for North America were calculated for the period from Jan. 1 to Feb. 1, 2017 and are shown in Fig. 6. Sharply increased sampling frequency is evident over much of Canada and over much of the eastern United States where V8 sampling frequency is near zero. For Asia, V8 and V9 sampling frequency maps were also calculated for the period from Jan. 1 to Feb.

1, 2017 and are shown in Fig. 7. Increased sampling frequency is apparent over much of the continent, particularly western China, northeastern China and Mongolia.



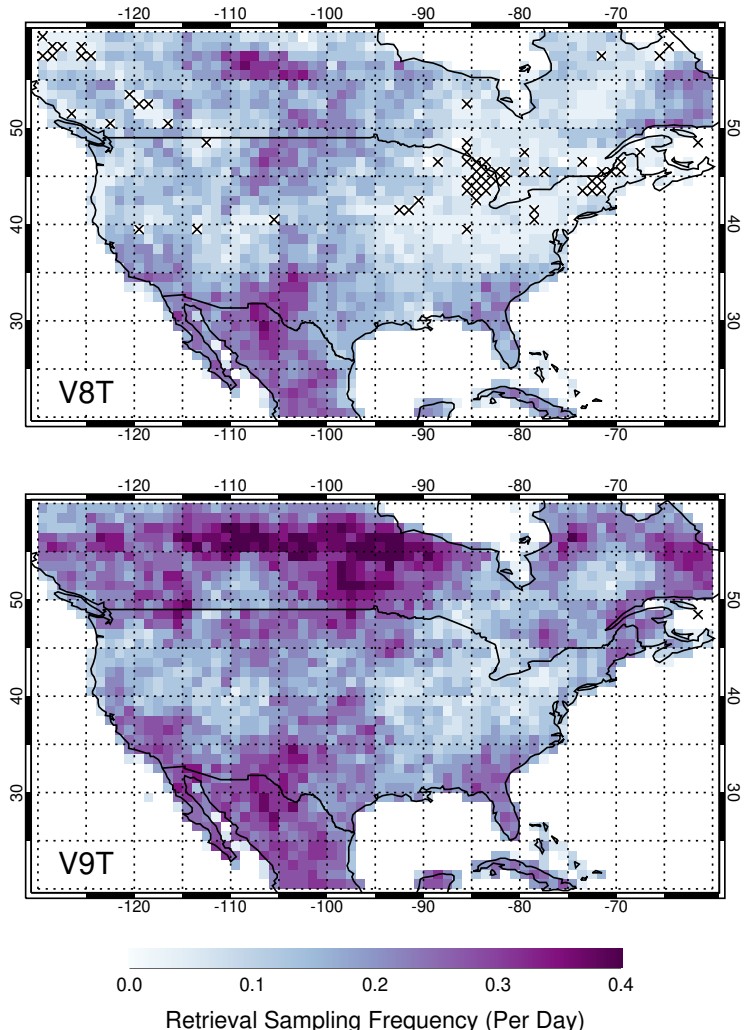

**Figure 6.** Comparison of maps of sampling frequency for North America for V8 and V9 TIR-only variants based on daytime/land retrievals for January, 2017. See caption to Fig. 5.

## 4.3  Level 3 Products

The beneficial effects of the cloud detection revisions are also readily apparent in the gridded MOPITT Level 3 monthly product, as shown in Fig. 8. The top row in this figure compares V8 and V9 TIR-NIR gridded monthly-mean daytime CO total column values for eastern China for January, 2010. Empty grid cell values, indicated in white, are much more common in the




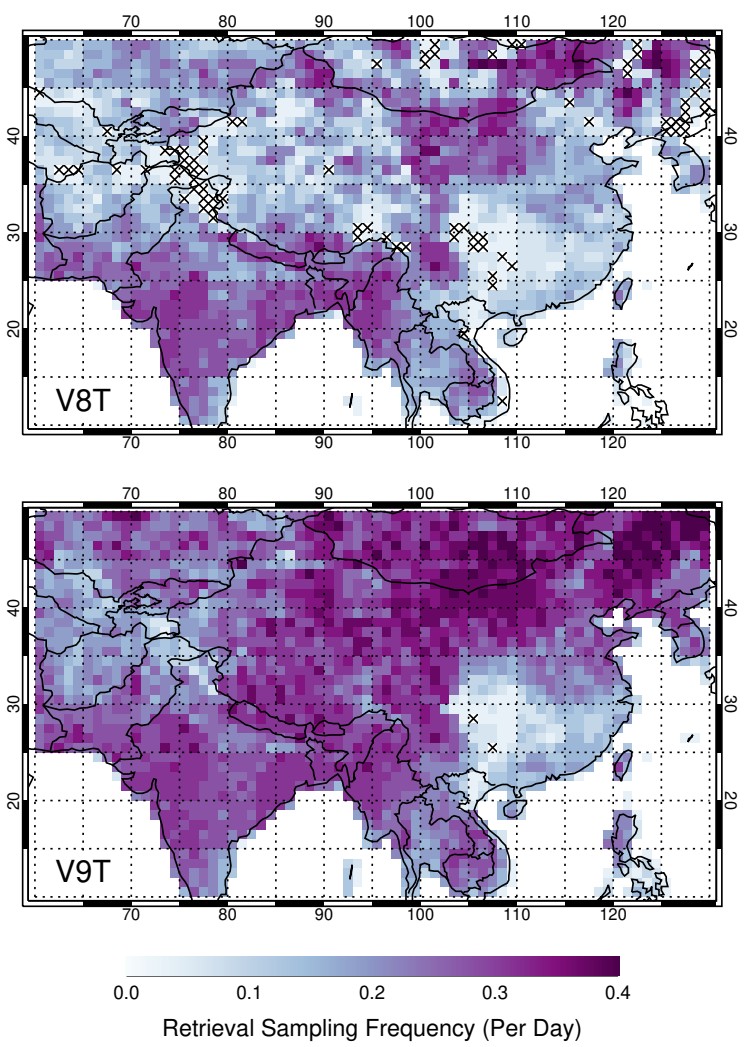

**Figure 7.** Comparison of maps of sampling frequency for East Asia for V8 and V9 TIR-only variants based on daytime/land retrievals for January, 2017. See caption to Fig. 5.

V8 product than in the V9 product. The bottom panel in the figure presents a map of the fractional difference derived from the top-row panels. This map demonstrates that over a heavily-polluted region such as the North China Plain, monthly-mean total column values in the V9 product may be larger than corresponding V8 values by 20% or more. This effect is due to the tendency of heavy aerosol loading to lead to the Uncertain outcome for the MODIS cloud mask, resulting in the exclusion of





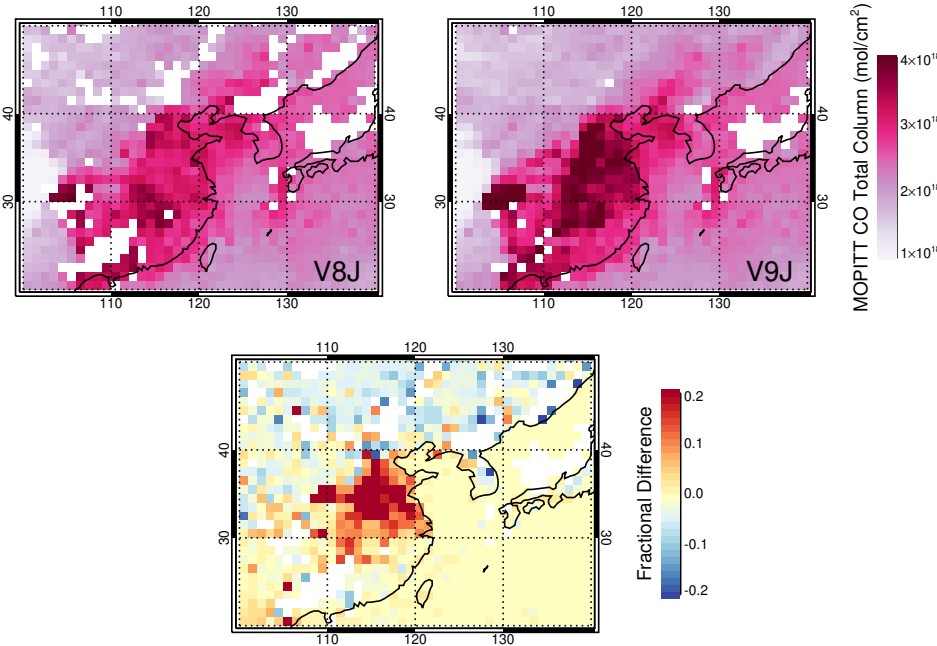

**Figure 8.** Comparison of V8 and V9 TIR-NIR maps of Level 3 monthly-mean CO total column (top) and corresponding fractional difference (bottom) for eastern China based on daytime retrievals for January, 2010.

such scenes in the MOPITT V8 product (Deeter et al., 2021). Thus, CO monthly-means in the V9 product should be more accurate than for V8 because retrievals are averaged over a wider and more complete range of pollution levels.

## 5   Conclusions

Various aspects of the MOPITT calibration methods and retrieval algorithm have been revised since the instrument became operational in 2000. For the most recently released Version 9 products, significant revisions were made to the NIR calibration

scheme and to the cloud detection algorithm. The new NIR calibration method was shown to reduce an apparent discontinuity in NIR-only retrievals for dates just before and just after the annual hot calibration/decontamination procedure. This revision should improve the temporal consistency of both the NIR-only and TIR-NIR products. The revised cloud detection algorithm allows retrievals in ambiguous situations (with respect to cloudiness) resulting in an increase in large-scale retrieval coverage over land of $\sim$ 30-40% compared to the V8 product. Validation results based on aircraft in-situ profiles indicate that V9 product

retrieval biases are typically in the range of $\pm5\%$ and are generally comparable to results for the V8 product.

The improved retrieval coverage and sampling frequency for V9 should add value to the MOPITT product in a wide variety of applications. For example, more frequent retrievals in CO source regions, such as the fire-prone Amazon Basin and heavily-





industrialized North China Plain, should lead to more accurate emissions estimates using inverse modeling methods. For visualizing CO distributions using monthly-mean maps, the new product is more statistically robust and has many fewer gaps

due to missing data. Moreover, heavily-polluted regions should be more accurately represented in such maps since the previous cloud detection algorithm tended to exclude the most heavily polluted scenes. Finally, the increased retrieval coverage should lead to better statistics when validating other satellite products.

*Data availability.*   The MOPITT Version 8 and Version 9 products are available from NASA through the Earthdata portal (https://earthdata.nasa.gov/) or directly from the ASDC archive (https://asdc.larc.nasa.gov/data/MOPITT/).





## Appendix A: Cloud index-subsetted validation results

NOAA V8 and V9 TIR-only validation results subsetted by cloud index value (1-6) are shown in Fig. A1 and are listed in Table
A1. The number of V8 and V9 retrievals within each subset are indicated in the figure legend and in the leftmost column of the
table. Corresponding results for the NIR-only and TIR-NIR products are presented in Figures A2 and A3 and Tables A2 and
A3. Cloud index values are defined in Section 2.3. A comparison of the numbers of V8 and V9 retrievals in Tables A1, A2, and
A3 indicates that the large majority of added retrievals in V9 (not present in the V8 product) are either assigned cloud index 2
('MODIS-clear, MOPITT-clear') or 6 ('MODIS-cloudy, MOPITT-clear').

For the V9 TIR-only results, retrieval biases cloud index subsets 2-6 fall in the range of ±5%. (The cloud index 1 sub-
set, composed of retrievals for which the MODIS cloud mask was unavailable, represents only about 1% of the entire set of
retrievals analyzed, and may not be statistically significant.) Corresponding bias ranges for the V9 NIR-only and TIR-NIR vari-
ants are ±2% and ±10%, respectively. In relation to the cloud index 2 subset ('MODIS-clear, MOPITT-clear') subset, which
represents the retrieval subset most confidently cloud-free, biases for the cloud index 6 subset ('MODIS-cloudy, MOPITT-
clear') are within 2% at all levels. Similarly, differences in the index 2 and index 6 subsets for the NIR-only and TIR-NIR
variants are within 1% and 3%, respectively. Thus, comparing retrieval biases for cloud index 2 and 6, it appears that the results
of the MODIS cloud mask test are not significant. However, the importance of the MODIS cloud mask test may be greater
in specific contexts not represented in the validation results, such as nighttime retrievals over land. Since bias differences as-
sociated with the different cloud index values are generally similar in magnitude to bias variations over the vertical profile,
validation results shown in Figures A1, A2, and A3 do not imply a clear benefit to filtering based on cloud index.

*Author contributions.* MD led the development and evaluation of the V9 product and wrote the manuscript. DM integrated the algorithm
revisions into the prototype and operational processing software and managed the data processing. GF implemented the revisions made to the
operational radiative transfer model. SMA managed the acquisition and processing of the in-situ datasets used for validation. MD, GF, DM,
JG, SMA, HW, DZ and JD participated in the development and testing of the revised NIR calibration scheme. RC, GD and KM provided
expertise with the in-situ datasets used for validation. All authors reviewed the manuscript.

*Competing interests.* The authors declare that they have no competing interests.

*Acknowledgements.* The NCAR MOPITT project is supported by the National Aeronautics and Space Administration (NASA) Earth Ob-
serving System (EOS) Program. The National Center for Atmospheric Research (NCAR) is sponsored by the National Science Foundation.
We also acknowledge the Canadian Space Agency who provided the MOPITT instrument and continue to support instrument operations.
In-situ datasets used for validation were provided by NOAA's Global Monitoring Laboratory and its partners, as well as participants in the
HIPPO, ATom and KORUS-AQ field campaigns.





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



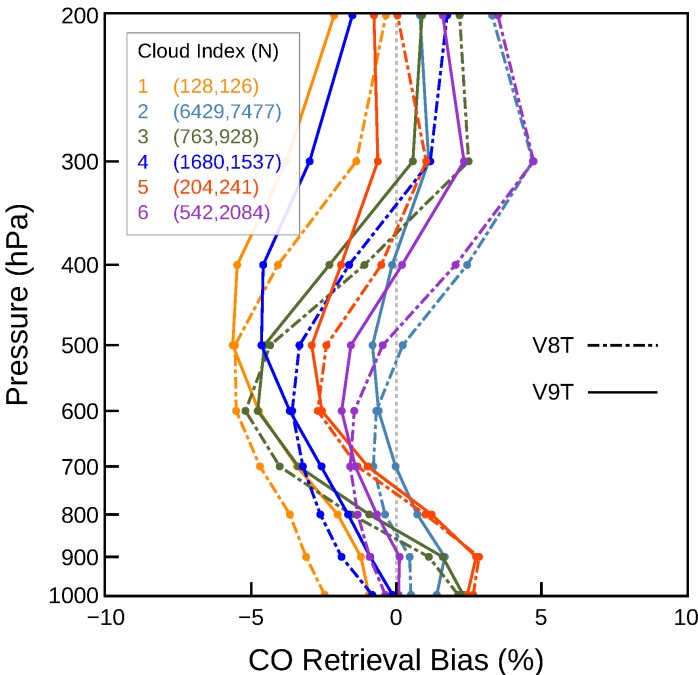

**Figure A1.** Cloud index-subsetted validation results for the V8 and V9 TIR-only variants using the NOAA profile set. Numbers in parentheses in the legend indicate the number of retrievals within the subset for the corresponding cloud index value for the V8 and V9 products.

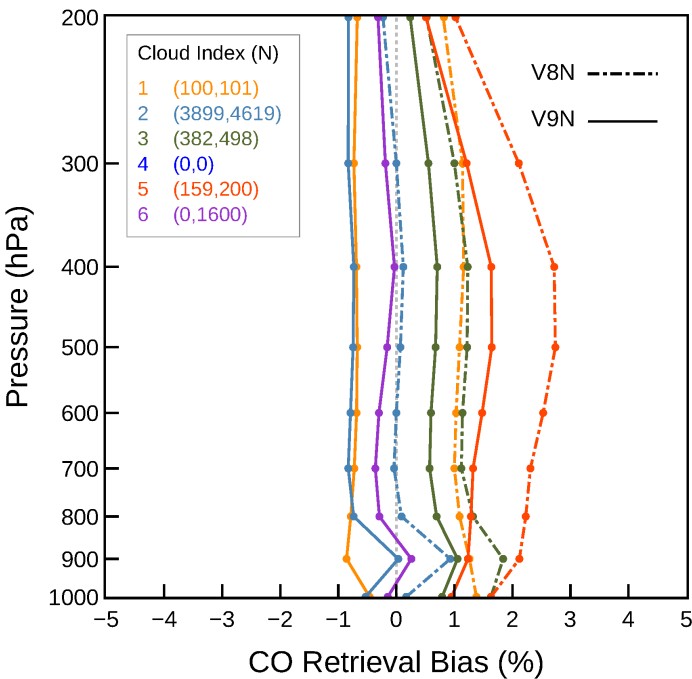

**Figure A2.** Cloud index-subsetted validation results for the V8 and V9 NIR-only variants using the NOAA profile set. Numbers in parentheses in the legend indicate the number of retrievals within the subset for the corresponding cloud index value for the V8 and V9 products.



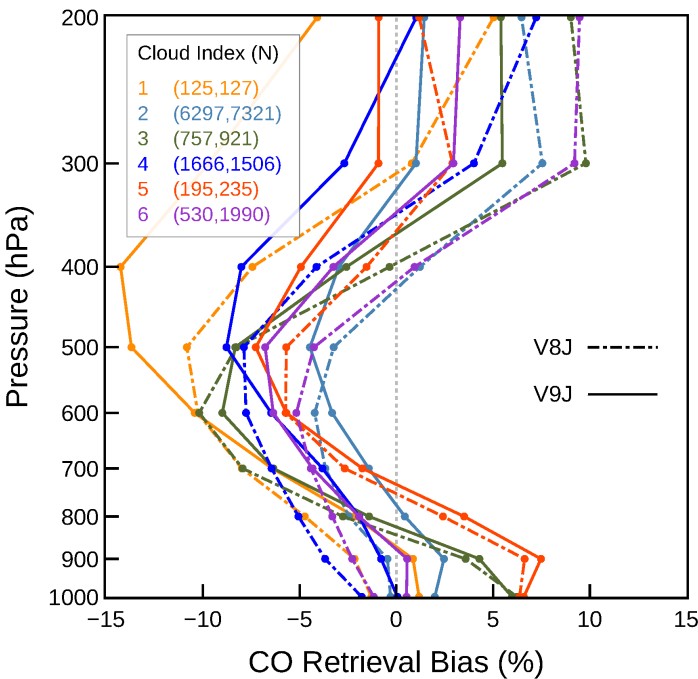

**Figure A3.** Cloud index-subsetted validation results for the V8 and V9 TIR-NIR variants using the NOAA profile set. Numbers in parentheses in the legend indicate the number of retrievals within the subset for the corresponding cloud index value for the V8 and V9 products.



**Table 1.** Radiance bias correction parameters used for processing MOPITT Version 9 retrieval products. See Section 2.2. $R_0$ is dimensionless. Units of $R_t$ and $R_w$ are day$^{-1}$ and (molecules/cm$^2$)$^{-1}$, respectively. Corresponding V8 values are indicated in parentheses only where they are different than V9 values.

|  | 5A | 5D | 6A | 6D | 7D |
|---|---|---|---|---|---|
| $R_0$ | 1.05970 | 1.04522 | 1.00000 | 0.99270 (0.99522) | 1.00955 (1.04959) |
| $R_t$ | 0.0 | 0.0 | 0.0 | $7.14\times10^{-7}$ ($9.6\times10^{-7}$) | $-2.0\times10^{-6}$ ($-1.18\times10^{-5}$) |
| $R_w$ | 0.0 | $-8.09\times10^{-27}$ | 0.0 | 0.0 | $-6.00\times10^{-25}$ |

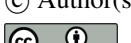



**Table 2.** Summarized validation results for V8 and V9 TIR-only (V8T and V9T), NIR-only (V8N and V9N) and TIR-NIR (V8J and V9J) variants based on in-situ data from NOAA aircraft validation sites. Total number of MOPITT retrievals used for validation are shown in parentheses in leftmost column. Bias and standard deviation (SD) statistics for the total column are given in units of molecules/cm$^2$. Bias and SD for retrieval levels are expressed in %. Total column drift values are provided in units of molecules/cm$^2$/yr. Drift for the retrieval levels is expressed in %/yr.

| | | Total Column | Surface | 800hPa | 600hPa | 400hPa | 200hPa |
|---|---|---|---|---|---|---|---|
| V8T | bias | $1.33 \times 10^{16}$ | 0.36 | -0.93 | -1.69 | 1.30 | 2.85 |
| (9746) | SD | $2.01 \times 10^{17}$ | 8.02 | 9.70 | 11.71 | 19.51 | 15.33 |
| | drift | $(-1.17 \pm 0.42) \times 10^{15}$ | $-0.022 \pm 0.017$ | $-0.030 \pm 0.021$ | $-0.059 \pm 0.025$ | $-0.102 \pm 0.041$ | $-0.079 \pm 0.032$ |
| V9T | bias | $9.69 \times 10^{15}$ | 1.05 | 0.05 | -1.59 | -0.88 | 0.61 |
| (12393) | SD | $2.16 \times 10^{17}$ | 7.79 | 9.70 | 12.35 | 21.22 | 15.76 |
| | drift | $(-1.52 \pm 0.41) \times 10^{15}$ | $-0.078 \pm 0.015$ | $-0.132 \pm 0.018$ | $-0.117 \pm 0.023$ | $-0.001 \pm 0.040$ | $0.075 \pm 0.030$ |
| V8N | bias | $1.96 \times 10^{16}$ | 0.37 | 0.29 | 0.20 | 0.32 | -0.10 |
| (4540) | SD | $2.46 \times 10^{17}$ | 10.97 | 11.21 | 10.86 | 12.12 | 8.35 |
| | drift | $(3.80 \pm 0.80) \times 10^{15}$ | $0.180 \pm 0.036$ | $0.211 \pm 0.037$ | $0.220 \pm 0.035$ | $0.240 \pm 0.039$ | $0.181 \pm 0.027$ |
| V9N | bias | $4.60 \times 10^{15}$ | -0.31 | -0.48 | -0.52 | -0.40 | -0.59 |
| (7018) | SD | $2.45 \times 10^{17}$ | 10.42 | 10.72 | 10.50 | 12.19 | 7.92 |
| | drift | $(3.27 \pm 0.63) \times 10^{15}$ | $0.149 \pm 0.027$ | $0.175 \pm 0.028$ | $0.181 \pm 0.027$ | $0.200 \pm 0.031$ | $0.143 \pm 0.020$ |
| V8J | bias | $1.82 \times 10^{16}$ | 0.04 | -2.90 | -5.47 | -0.00 | 6.84 |
| (9570) | SD | $2.29 \times 10^{17}$ | 17.16 | 17.37 | 14.48 | 24.99 | 27.34 |
| | drift | $(-2.27 \pm 4.88) \times 10^{14}$ | $0.097 \pm 0.037$ | $-0.002 \pm 0.037$ | $-0.084 \pm 0.031$ | $-0.108 \pm 0.053$ | $0.074 \pm 0.058$ |
| V9J | bias | $1.60 \times 10^{16}$ | 1.90 | -0.35 | -4.76 | -3.76 | 1.90 |
| (12100) | SD | $2.42 \times 10^{17}$ | 17.81 | 17.37 | 14.76 | 27.34 | 26.67 |
| | drift | $(-3.16 \pm 4.58) \times 10^{14}$ | $-0.028 \pm 0.034$ | $-0.204 \pm 0.033$ | $-0.186 \pm 0.028$ | $0.073 \pm 0.052$ | $0.365 \pm 0.050$ |





**Table 3.** Summarized validation results for V8 and V9 TIR-only (V8T and V9T) variants based on in-situ data from the HIPPO field campaign. See caption to Table 2.

|  |  | Total Column | Surface | 800hPa | 600hPa | 400hPa | 200hPa |
|---|---|---|---|---|---|---|---|
| V8T | bias | $4.77\times10^{15}$ | 5.51 | 2.37 | -3.40 | -3.93 | -0.31 |
| (10547) | SD | $1.43\times10^{17}$ | 12.18 | 11.47 | 13.77 | 16.50 | 13.98 |
| V9T | bias | $-2.06\times10^{15}$ | 5.56 | 3.13 | -3.26 | -5.86 | -2.23 |
| (11613) | SD | $1.53\times10^{17}$ | 11.42 | 11.74 | 13.84 | 17.48 | 14.61 |

**Table 4.** Summarized validation results for V8 and V9 TIR-only (V8T and V9T) variants based on in-situ data from the AToM field campaign. See caption to Table 2.

|  |  | Total Column | Surface | 800hPa | 600hPa | 400hPa | 200hPa |
|---|---|---|---|---|---|---|---|
| V8T | bias | $-1.40\times10^{16}$ | -0.47 | -2.43 | -4.42 | -1.03 | 3.30 |
| (10512) | SD | $1.73\times10^{17}$ | 5.84 | 8.78 | 11.68 | 16.35 | 14.51 |
| V9T | bias | $-1.22\times10^{16}$ | 0.00 | -1.69 | -4.04 | -1.79 | 2.11 |
| (11242) | SD | $1.87\times10^{17}$ | 5.93 | 8.77 | 12.02 | 17.71 | 14.97 |



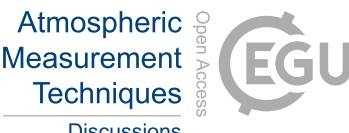

**Table 5.** Summarized validation results for V8 and V9 TIR-only (V8T and V9T), NIR-only (V8N and V9N) and TIR-NIR (V8J and V9J) variants based on in-situ data from the KORUS-AQ field campaign. See caption to Table 2.

| | | Total Column | Surface | 800hPa | 600hPa | 400hPa | 200hPa |
|---|---|---|---|---|---|---|---|
| V8T | bias | $-1.20\times10^{16}$ | -0.16 | -2.90 | -2.55 | 1.83 | 3.92 |
| (217) | SD | $2.19\times10^{17}$ | 9.85 | 8.88 | 11.14 | 21.46 | 17.33 |
| V9T | bias | $1.91\times10^{16}$ | 0.71 | -1.71 | -2.48 | 0.35 | 3.24 |
| (280) | SD | $3.44\times10^{17}$ | 9.31 | 9.47 | 13.73 | 27.56 | 23.88 |
| V8N | bias | $9.15\times10^{14}$ | 0.22 | 0.10 | -0.02 | -0.01 | -0.04 |
| (130) | SD | $2.25\times10^{17}$ | 9.47 | 7.14 | 5.76 | 5.63 | 3.78 |
| V9N | bias | $-2.96\times10^{16}$ | -1.22 | -0.99 | -0.88 | -0.85 | -0.65 |
| (185) | SD | $2.18\times10^{17}$ | 9.21 | 7.01 | 5.78 | 5.69 | 3.81 |
| V8J | bias | $-1.80\times10^{16}$ | -2.25 | -6.78 | -5.89 | 3.52 | 11.50 |
| (215) | SD | $2.23\times10^{17}$ | 20.29 | 14.17 | 11.26 | 27.32 | 31.31 |
| V9J | bias | $1.84\times10^{16}$ | 0.39 | -3.75 | -6.47 | -2.08 | 6.56 |
| (272) | SD | $3.37\times10^{17}$ | 20.44 | 15.07 | 15.28 | 35.25 | 36.98 |




**Table A1.** Summarized validation results for V8 and V9 TIR-only (V8T and V9T), subsetted by cloud index values and based on in-situ data from the NOAA aircraft stations. See caption to Table 2.

| | | Total Column | Surface | 800hPa | 600hPa | 400hPa | 200hPa |
|---|---|---|---|---|---|---|---|
| V8T-1 | bias | $-4.47\times10^{16}$ | -2.47 | -3.67 | -5.52 | -4.08 | -0.36 |
| (128) | SD | $2.66\times10^{17}$ | 7.41 | 9.68 | 13.97 | 26.15 | 20.62 |
| V8T-2 | bias | $2.56\times10^{16}$ | 0.50 | -0.39 | -0.69 | 2.44 | 3.30 |
| (6429) | SD | $1.98\times10^{17}$ | 8.30 | 9.85 | 11.10 | 18.93 | 14.73 |
| V8T-3 | bias | $-1.50\times10^{16}$ | 2.13 | -1.55 | -5.20 | -1.10 | 2.18 |
| (763) | SD | $1.77\times10^{17}$ | 10.12 | 7.50 | 10.29 | 16.19 | 10.67 |
| V8T-4 | bias | $-2.12\times10^{16}$ | -0.83 | -2.62 | -3.59 | -1.63 | 1.75 |
| (1680) | SD | $1.65\times10^{17}$ | 5.86 | 10.01 | 12.55 | 18.70 | 17.04 |
| V8T-5 | bias | $2.80\times10^{16}$ | 2.65 | 1.00 | -2.71 | -0.52 | 0.04 |
| (204) | SD | $3.29\times10^{17}$ | 8.68 | 10.05 | 17.45 | 29.21 | 11.62 |
| V8T-6 | bias | $2.10\times10^{16}$ | -0.38 | -1.33 | -1.45 | 2.04 | 3.50 |
| (542) | SD | $2.64\times10^{17}$ | 5.73 | 8.69 | 12.83 | 24.60 | 20.86 |
| V9T-1 | bias | $-3.93\times10^{16}$ | -0.94 | -2.02 | -4.74 | -5.49 | -2.14 |
| (126) | SD | $2.44\times10^{17}$ | 7.05 | 8.59 | 11.91 | 24.22 | 19.29 |
| V9T-2 | bias | $1.81\times10^{16}$ | 1.40 | 0.72 | -0.61 | -0.14 | 0.81 |
| (7477) | SD | $1.94\times10^{17}$ | 8.09 | 9.72 | 11.03 | 19.11 | 14.17 |
| V9T-3 | bias | $-1.52\times10^{16}$ | 2.30 | -0.94 | -4.78 | -2.31 | 0.89 |
| (928) | SD | $1.80\times10^{17}$ | 9.60 | 7.44 | 10.69 | 17.32 | 11.22 |
| V9T-4 | bias | $-3.13\times10^{16}$ | -0.13 | -1.66 | -3.67 | -4.59 | -1.52 |
| (1537) | SD | $1.67\times10^{17}$ | 5.88 | 10.52 | 13.21 | 19.49 | 16.93 |
| V9T-5 | bias | $1.40\times10^{16}$ | 2.44 | 1.22 | -2.56 | -1.91 | -0.77 |
| (241) | SD | $2.83\times10^{17}$ | 8.26 | 10.45 | 17.01 | 27.23 | 10.78 |
| V9T-6 | bias | $2.26\times10^{16}$ | 0.07 | -0.67 | -1.88 | 0.20 | 1.60 |
| (2084) | SD | $3.06\times10^{17}$ | 6.72 | 9.63 | 15.45 | 28.61 | 21.22 |





**Table A2.** Summarized validation results for V8 and V9 NIR-only (V8N and V9N), subsetted by cloud index values and based on in-situ data from the NOAA aircraft stations. See caption to Table 2.

| | | Total Column | Surface | 800hPa | 600hPa | 400hPa | 200hPa |
|---|---|---|---|---|---|---|---|
| V8N-1 | bias | $3.05 \times 10^{16}$ | 1.38 | 1.09 | 1.03 | 1.16 | 0.82 |
| (100) | SD | $1.77 \times 10^{17}$ | 9.05 | 8.91 | 8.19 | 8.91 | 7.32 |
| V8N-2 | bias | $1.52 \times 10^{16}$ | 0.17 | 0.09 | -0.00 | 0.12 | -0.23 |
| (3899) | SD | $2.39 \times 10^{17}$ | 10.85 | 11.02 | 10.62 | 11.98 | 8.48 |
| V8N-3 | bias | $4.63 \times 10^{16}$ | 1.63 | 1.32 | 1.14 | 1.23 | 0.52 |
| (382) | SD | $3.06 \times 10^{17}$ | 12.78 | 12.47 | 11.92 | 12.53 | 7.49 |
| V8N-5 | bias | $5.55 \times 10^{16}$ | 1.63 | 2.23 | 2.53 | 2.72 | 1.02 |
| (159) | SD | $2.75 \times 10^{17}$ | 10.03 | 13.33 | 14.72 | 15.66 | 7.74 |
| V9N-1 | bias | $-5.69 \times 10^{15}$ | -0.46 | -0.79 | -0.69 | -0.69 | -0.67 |
| (101) | SD | $1.67 \times 10^{17}$ | 8.70 | 8.61 | 7.89 | 8.56 | 7.07 |
| V9N-2 | bias | $-1.02 \times 10^{15}$ | -0.53 | -0.73 | -0.79 | -0.73 | -0.82 |
| (4619) | SD | $2.29 \times 10^{17}$ | 10.44 | 10.59 | 10.21 | 11.44 | 8.09 |
| V9N-3 | bias | $2.79 \times 10^{16}$ | 0.79 | 0.69 | 0.60 | 0.70 | 0.24 |
| (498) | SD | $2.76 \times 10^{17}$ | 11.47 | 11.73 | 11.51 | 12.30 | 7.38 |
| V9N-5 | bias | $3.43 \times 10^{16}$ | 0.95 | 1.28 | 1.48 | 1.64 | 0.51 |
| (200) | SD | $2.57 \times 10^{17}$ | 9.56 | 12.69 | 13.96 | 14.85 | 7.42 |
| V9N-6 | bias | $1.06 \times 10^{16}$ | -0.15 | -0.29 | -0.30 | -0.03 | -0.32 |
| (1600) | SD | $2.79 \times 10^{17}$ | 10.20 | 10.56 | 10.62 | 13.95 | 7.68 |





**Table A3.** Summarized validation results for V8 and V9 TIR-NIR (V8J and V9J), subsetted by cloud index values and based on in-situ data from the NOAA aircraft stations. See caption to Table 2.

|  |  | Total Column | Surface | 800hPa | 600hPa | 400hPa | 200hPa |
|---|---|---|---|---|---|---|---|
| V8J-1 | bias | $-3.07\times10^{16}$ | -1.29 | -4.73 | -10.26 | -7.44 | 5.03 |
| (125) | SD | $2.43\times10^{17}$ | 17.64 | 15.93 | 13.13 | 29.62 | 35.25 |
| V8J-2 | bias | $2.63\times10^{16}$ | -0.30 | -2.45 | -4.22 | 1.22 | 6.47 |
| (6297) | SD | $2.08\times10^{17}$ | 17.76 | 17.46 | 13.58 | 24.87 | 26.38 |
| V8J-3 | bias | $3.38\times10^{16}$ | 6.24 | -2.76 | -10.19 | -0.35 | 9.02 |
| (757) | SD | $3.22\times10^{17}$ | 22.19 | 16.11 | 14.06 | 21.91 | 22.84 |
| V8J-4 | bias | $-2.44\times10^{16}$ | -1.79 | -5.06 | -7.77 | -4.13 | 7.23 |
| (1666) | SD | $1.91\times10^{17}$ | 11.69 | 18.36 | 17.01 | 22.93 | 29.63 |
| V8J-5 | bias | $4.28\times10^{16}$ | 6.37 | 2.41 | -5.72 | -1.54 | 1.18 |
| (195) | SD | $2.74\times10^{17}$ | 18.41 | 16.19 | 15.95 | 31.27 | 16.81 |
| V8J-6 | bias | $3.36\times10^{16}$ | -1.17 | -3.31 | -5.17 | 0.94 | 9.47 |
| (530) | SD | $3.44\times10^{17}$ | 11.91 | 14.87 | 14.32 | 30.66 | 36.13 |
| V9J-1 | bias | $-5.99\times10^{16}$ | 1.18 | -2.17 | -10.42 | -14.25 | -4.09 |
| (127) | SD | $2.28\times10^{17}$ | 17.30 | 16.11 | 15.75 | 30.53 | 31.27 |
| V9J-2 | bias | $2.38\times10^{16}$ | 1.99 | 0.44 | -3.32 | -2.98 | 1.44 |
| (7321) | SD | $2.18\times10^{17}$ | 17.81 | 17.46 | 13.39 | 25.76 | 25.34 |
| V9J-3 | bias | $2.28\times10^{16}$ | 5.98 | -1.41 | -9.00 | -2.58 | 5.40 |
| (921) | SD | $2.98\times10^{17}$ | 20.72 | 15.58 | 14.06 | 23.83 | 21.64 |
| V9J-4 | bias | $-3.04\times10^{16}$ | 0.06 | -1.92 | -6.47 | -8.01 | 1.03 |
| (1506) | SD | $1.89\times10^{17}$ | 11.52 | 18.71 | 16.86 | 23.89 | 27.99 |
| V9J-5 | bias | $2.94\times10^{16}$ | 6.59 | 3.50 | -5.69 | -4.93 | -0.92 |
| (235) | SD | $2.81\times10^{17}$ | 20.22 | 17.89 | 18.11 | 30.98 | 17.04 |
| V9J-6 | bias | $2.20\times10^{16}$ | 0.53 | -1.93 | -6.36 | -3.26 | 3.30 |
| (1990) | SD | $3.12\times10^{17}$ | 19.43 | 16.59 | 16.84 | 34.81 | 32.32 |