# Peer review of "The MOPITT Version 9 CO Product: Sampling Enhancements and Validation"

_Atmospheric Measurement Techniques, 2021_

## Referee Comment (RC2)

[referee-annotated manuscript omitted]

---

## Author Comment (AC1)

Authors' response to reviews of "The MOPITT Version 9 CO Product: Sampling Enhancements and Validation" by M. Deeter et al.

Original reviewer's comments in blue. Authors' responses in black.

**Replies to Comments of Reviewer #1**

In this paper the authors present new changes implemented in the MOPITT V9 algorithm and how the results before and after the changes compare with aircraft measurements. The latest algorithm uses a calibration method that produces more negligible discontinuities and processes more scenes, especially those previously assumed to be too cloudy or polluted. Overall I found the paper to be well written, insightful, and clear. The improved coverage mentioned in Section 4 looks like a very nice result for the community.

I recommend publication with minor revisions. My biggest overall comment is often a sentence or two of additional detail is needed for the rest of the community. Sixteen of the thirty-one references were led by authors on this paper, which to me highlights the immense combined expertise of the authors which those in the rest of the scientific community do not necessarily have.

I wish the authors a safe and happy holiday season.

Response to General Comments: We appreciate the reviewer's thorough review and have made numerous revisions to the text.

Specific comments

S1 - P2, L24: Is the top level 100 hPa or 50 hPa? Please clarify. If the retrieval algorithm does not go to the top of the atmosphere, please specify what is used for the top layer up to the TOA.

Authors' reply: The following text has been added for clarification: "For the topmost MOPITT retrieval level at 100 hPa, the uniform-VMR layer extends from 100 hPa to 50 hPa. Assumed VMR values in the layer from 50 hPa to TOA are based on the CAM-chem model climatology and are fixed."

S2 - P2, L30: To save the reader from looking up other papers, could you provide an extra sentence or two describing the prior? E.g., spatiotemporal resolution and are there inter as well as intra annual trends?

Authors' reply: The following text has been added for clarification: "MOPITT a priori log(VMR) profiles vary by month, but do not vary from year to year; this simplifies the interpretation of long-term trends in the data. Model-based climatologies used to generate the a priori are gridded at one degree (lat/lon) horizontal resolution and monthly temporal resolution. Spatial and temporal interpolation are used to generate a priori values at each specific observation location and day."

S3 - P2, L38: Does low SNR preclude use of 7A? It would be helpful to mention the cooler failure and other bands not being designed to quantify CO so those new to MOPITT don't wonder why the numbering starts at 5.

Authors' reply:  The Channel 7 Average radiance is not used in the retrievals because the available information in that radiance is redundant with information from the other thermal-channel radiances which are used.  Experiments conducted for earlier MOPITT product versions (unpublished) showed no benefits to inclusion of the Channel 7 Average radiance.  The following text has been added regarding the cooler failure: "Radiometers on MOPITT corresponding to channels 1-4 became inoperative in 2001 due to the failure of one of two coolers."

S4 - P2, L38: "J" often shows up in figures for the TIR-NIR retrievals, please define here. I think it means "joint" but I am not finding that word in this paper.

Authors' reply:  The word 'joint' now prefaces the description of the TIR-NIR product in Section 1: "Unique 'multi-spectral' or 'joint' TIR-NIR retrievals exploit the 5A, 5D, 7D, 6D, and 6A radiances." In addition, the first sentence of the third paragraph in Section 3.1 now reads: "Validation results derived from the NOAA aircraft flask samples for the V8 and V9 TIR-only (V8T and V9T), NIR-only (V8N and V9N) and joint TIR-NIR (V8J and V9J) variants are compared in Fig. 2."

S5 - P3, L66: Why March every year? Is this the only time that works for Terra, or is it just arbitrary? It seems like this could cause sampling errors.

Authors' reply:  MOPITT instrument maintenance operations are required annually.  The month of March was selected by the instrument PI for such operations since CO variability (due to fires, for example) is often weaker for March than for other months and therefore the pause in MOPITT observations should have a minimal impact on users.

S6 - P5, L99-102: I don't fully understand how these 2 sentences relate. Were p and T calculated but not used to update optical depth before?

Authors' reply:  The following text has been added for clarity: "For V9, operational modeling of the MOPITT Pressure Modulation Cell (PMC) radiances (7A and 7D) now also includes monthly updated values for the cell number density. The optical depth is calculated as the product of the cross-section, number density and cell length. Monthly variations in cell pressure (P) and temperature (T) affect the number density, which is proportional to P/T. This dependency is now explicitly represented in V9. This correction removes a small but slowly growing bias in the 7D PMC radiance (~0% in 2006, ~3% in 2018) which is large enough to introduce a non-negligible long-term trend in CO retrieval bias."

S7 - P5, L104: From Deeter et al., 2014 it seems these are scalar factors for 5A, 5D, and 7D, correct? If so, could you add the values to this paragraph? Or do these factors get rolled up with R_0 already listed in Table 1?

Authors' reply:  For earlier MOPITT products, a single fixed radiance correction factor was used for each radiance used by the retrieval algorithm. For V9, radiance correction factors depend also on time and water vapor amounts, as defined in Eq. 1. The values listed in Table 1 completely describe the radiance bias correction parameters used for V9 processing.

S8 - P6, L131: Why does the radiative transfer model only use clear-sky conditions? Is it to save on computational expense? While computational power has increased over the past 21-years, I presume the model has also become more complex which precludes just trying to process all soundings and filtering later?

Authors' reply:  There are several issues preventing the retrieval of CO in cloudy conditions.  The most significant problem is simply that the MOPITT observed radiances do not contain sufficient information to simultaneously retrieve CO concentrations and the many cloud parameters (such as cloud fraction, cloudtop height, and cloud optical properties) which affect the radiances.  Ignoring the effects of clouds completely would severely degrade the quality of the MOPITT product.

S9 - P7, L174: Marey et al., 2021 (https://amt.copernicus.org/preprints/amt-2021-112/) have also looked into the MOPITT cloud filtering method and recommended reconsideration of processing land scenes with low clouds. For completeness that study should also be referenced, even if the suggestion was unused.

Authors' reply: A reference to that paper (now titled 'Analysis of improvements in MOPITT observational coverage over Canada') has been added to Section 4.2.

S10 - P8, L208: Consider changing "retrieval error" to just "error" to make it more generic and account for all the various sources of error/noise (e.g., Rodgers, 2000 eq. 3.16). E.g., there is uncertainty on the in-situ measurements.

Authors' reply:  The original text in the first paragraph of Section 3, along with Eq. 2, did not make it clear that the equation was included only to describe how 'simulated' MOPITT retrievals were calculated using in-situ profiles and the MOPITT averaging kernels.  We use the term 'retrieval error' to refer specifically to observed differences between retrieved quantities and corresponding simulated quantities. We have clarified this point with revised text and an additional equation where 'retrieval error' is explicitly defined.

S11 - P8, Eq2: I prefer to see an explicit generic error term, like in Rodgers and Connor 2003 (Eq. 3 therein) and then a statement afterwards saying the error term is neglected rather than just neglecting the error term from the beginning. Given that the error actually seems to be the major focus of Section 3, it makes more sense to explicitly include it. As written, Eq. 2 is only correct in the context of the accompanying sentence, not as a stand-alone.

Authors' reply: See reply to previous comment (immediately above).

S12 - P9, L230: Is this 50 km from the center, the nearest edge, or the furthest edge of the MOPITT sounding? If the soundings are about 22 km wide this could make a difference.

Authors' reply:  Geolocation data (latitude and longitude) for the MOPITT retrievals refer to the center of the footprint at the surface, as is customary. The first sentence of this paragraph now reads: "… a maximum separation of 50 km was employed (relative to the center of the MOPITT 22 by 22 km footprint) … "

S13 - P9, L233: I'm trying to understand how this is not circular since the CAM-chem model is also used for the a priori. Have you tried looking only at results for where measurements are available?

Authors' reply: The lack of in-situ data at high altitudes (e.g., pressures less than ~ 200 hPa) should not typically cause significant error because of the weak sensitivity of MOPITT radiances to CO in that part of the atmosphere. This issue will be examined quantitatively in a soon-to-be-submitted MOPITT validation paper exploiting in-situ profiles from the NOAA AirCore program. These in-situ profiles reach much higher altitudes than traditional aircraft-based profiles.

S14 - P10, Fig 2: I'd like to see estimates of uncertainty. Maybe error bars for all 6 profiles would be too cluttery, but a representative 1 standard error of the mean region (on TIR V9?) could perhaps be used since standard deviations are too large.

Authors' reply:  While uncertainty values resulting from the optimal estimation retrieval algorithm are provided with each MOPITT retrieval, the focus of this manuscript (and previous MOPITT validation papers) is on retrieval bias. As explained in the last paragraph of Section 3.1, the standard deviation values listed in Table 2 should not be interpreted as being equivalent to random retrieval error.  To avoid misinterpretation, we have chosen to list the standard deviation values in Table 2 but not plot them as error bars in the figures.

S15 - P10, Fig 2 (and 3 and A1-A3): Should these points be shifted up 50 hPa? For example, if the 900 hPa points really represent the layer average from 900-800 hPa then 850 hPa would be a more appropriate place to put the marker. If the values are just being plotted as representative indices then it does not make sense to show on a log-scale.

Authors' reply:  As specified in the MOPITT Level 2 data files, CO profiles are reported on a ten-level grid of pressure levels (surface, 900 hPa, ...).  The y-axis (pressure) values for the data in Fig. 2 correspond exactly to the retrieval levels in this grid. Altering this correspondence would create an inconsistency which would confuse many readers.  The relationship between retrieval levels and layers is described in Section 1 and in previous MOPITT publications (including the User's Guides).

S16 - P10, Sec 3.1: Are there any correction factors or attempts to correct for these biases and drifts in the products delivered through Earthportal or ASDC? If not, please comment in the paper on whether you recommend end users should attempt to include their own corrections based on these results or if data can be used as-is. Same comment for Sec 3.3.

Authors' reply: We do not recommend that users attempt to correct for retrieval biases in the MOPITT products available to users. This is now stated at the end of the first paragraph of Section 3: "While the validation results reported below are useful for estimating the magnitude of expected retrieval bias and drift, they should not be used as the basis for applying ad-hoc corrections to the MOPITT data."

S17 - Fig. 4: (Comment) These y-values appear to be in the correct locations. Gridlines could help the reader.

Authors' reply: Horizontal gridlines have been added every 30 degrees of latitude.

S18 - P19, Data availability: Please include how other datasets can be accessed as well including the various in situ profiles, and CAM-chem model results.

Authors' reply: The 'Data Availability' section has been updated to include URLs for the CO in-situ data repositories. An effort will also be made to post the CAM-chem CO climatology (used as the MOPITT a priori) on the MOPITT website.

S19 - Tables 2 and throughout: Please explicitly state the range for the levels. Presumably 800hPa is actually 800-700hPa (rather than say 900-800 hPa or 850-750 hPa). If there is not enough space in the column headers a short description and example could be added to the table caption of e.g., Table 2.

Authors' reply: See reply above to comment 'S15 - P10, Fig 2.' This change would create an inconsistency with the retrieval grid specified in the MOPITT data files and confuse readers. The relationship between retrieval levels and layers is clearly explained in Section 1.

Technical comments

T1 - P2, L19: "continuously" – "continually" may be a better word choice

Authors' reply:  Done.

T2 - P4, Fig 1: mean -> daily-mean

Authors' reply: Done.

T3 - P5, L99: Is MOPFAS an acronym? If so please add it here.

Authors' reply:  MOPFAS is the historical name given to the MOPITT operational radiative transfer model.

T4 - P6, L133: the MODIS -> the Terra MODIS… (to indicate they are on the same spacecraft)

Authors' reply: Done.

T5 - P7, L176: MOPITT observed to modeled radiance ratio…

Authors' reply: Done.

T6 - P11, L292: "below" -> "herein"

Authors' reply: Done.

T7 - Hyphens used in ranges should be en dashes throughout (presumably will be corrected anyways during copy editing)

Authors' reply: We trust the journal's editing staff will make any necessary formatting changes prior to publication.

Optional

O1 - P8, Section 3: I personally dislike the use of the word "validation" in this context, as it has the connotation of confirming something already presumed to be correct. However, there will almost certainly be a V10 product at some point, which I anticipate will have smaller errors and be more correct. Then V9 will be less-valid, though perhaps not invalid…I personally prefer terms like "comparison" or "intercomparison" (as in Rodgers and Connor, 2003), but can go either way as I realize much of the literature has unfortunately gone down the "validation" route in recent years.

Authors' reply:  We believe our usage of 'validation' is consistent with the remote sensing literature when comparing remote sensing products to a well-defined standard (such as calibrated flask samples). As used by Rodgers and others, the word 'intercomparison' is often used when comparing two remote sensing products where neither of the products can be considered 'truth.'

O2 - P10, Fig. 2: The authors may just consider turning on the grid in the plotting tool rather than the dashed vertical line at 0% which I temporarily thought was a profile. X ticks on the upper axis may also enhance clarity.

Authors' reply:  Both the linetype and color of the zero-reference line are different than for the plotted data and should therefore be visually distinct. Tick marks have been added to the top axis.

O3 - P10, Fig 2, 3, 4, A1-A3: More of just a note, but the connected lines are not representative of error profiles because values are bin averages and thus layer averages are not preserved on interpolation (e.g., Delhez 2002, doi: 10.1016/S0893-9659(02)00139-8). I am assuming their purpose is just to guide the eye, which is okay and to me preferred over unconnected dots.

---

## Author Comment (AC2)

Authors' response to reviews of "The MOPITT Version 9 CO Product: Sampling Enhancements and Validation" by M. Deeter et al.

Original reviewer's comments in blue. Authors' responses in black.

**Replies to Comments of Reviewer #2**

This paper describes the MOPITT V9 products, and compare the calibration results to V8. In general, the paper is very clear and focusses on the effect of modifying the cloud mask. This leads to a larger number of scenes that pass the MODIS/MOPITT filter criteria, specifically in cases in which heavy aerosol loading is present in the boundary layer. Using a number of examples, effects are clearly illustrated.

I will upload an annotated pdf, in which I made some small suggestions that might further improve this excellent paper. Specifically, while figure 4 focusses on the TIR product (only small differences in zonal mean compared to V8), Figure 8 shows results of the TIR-NIR product that show substantial increases. This hints to improved sensitivity to boundary layer pollution, a subject that is not fully exploited in the text. Likewise, Figures 6 and 7 show distinct increases in sampling frequency at higher latitudes. Here, it might be instructive to provide more insight in the physical reasons for this phenomenon. While boundary layer aerosols are mentioned as possible reason, the widespread enhancement in sampling frequency over Canada in Jan 2017 has likely other reasons.

Apart from that I am really satisfied with this paper.

Response to General Comments: We appreciate the reviewer's comments. The new cloud detection method does not affect retrieval sensitivity to CO in the boundary layer (in any particular retrieval), but it does recover significantly more retrievals in highly polluted conditions (e.g., in the North China Plain). As illustrated in the 2021 RSE paper where the new cloud detection method was first described, improved sampling for V9 is the result of both (1) added scenes where extreme pollution confused the MODIS cloud mask and (2) added scenes where possible clouds in the MOPITT field of view (e.g., low clouds or very thin clouds) produce little if any radiative effect. A reference has been added in Section 4.2 to a recent publication by Marey et al. where the added retrievals over Canada were primarily traced to scenes with low clouds.

*Responses to comments in annotated pdf:*

p. 2, l. 27: I do not think this is "introduction".

Authors' reply: Since we assume some readers will not have used prior MOPITT products, the introduction seems like an appropriate place to briefly describe the salient features of the retrieval algorithm, including the a priori.

p. 2, l. 42: I think "are" is better.

Authors' reply: '… should be relevant … ' has been replaced by '… will be relevant … '

p. 9, l. 231: please add the MOPITT foot print area for completeness.

Authors' reply: The MOPITT footprint (22 by 22 km) is now mentioned in the second paragraph of Section 3.1.

 Here it might be good to mention that the "extra" measurements (due to less strict cloud filter) did not severely influence the bias. But the question is really how many more profiles are included in the evaluation? Or is it based still on exactly the same valid comparisons?

Authors' reply:  Since there were two other algorithm changes made for V9 (the revised NIR calibration method and the radiative transfer model), the V8/V9 comparisons presented in Section 3.1 are not solely related to the revised cloud detection method.  The significant increase in the number of retrievals used to validate V9 relative to V8 can be seen in the numbers in the leftmost column of Tables 2-5 and A1.

p. 12, Fig. 3: Maybe good to explain why you do not compare the NIR and TIR-NIR for ATom and HIPPO.

Authors' reply: The following sentence has been added at the end of the first paragraph of Section 3.3: "Since MOPITT retrievals over ocean are based solely on TIR radiances, validation results presented below for the HIPPO and ATom campaigns (which mainly produced over-ocean observations) are limited to the TIR-only variant."

p. 13, last sentence of Section 4.1: mm, this sentence is too vague. Please be concrete. such as means are within ...%.

Authors' reply: The following sentence has been added before the final sentence in that paragraph: "V8T and V9T zonal means are within 2% at most latitude bands."  Also, in the preceding sentence, the phrase 'nearly negligible' has been replaced with 'very weak'.

p. 15 l. 362: not clear from the text why a different sampling period is chosen...Is this based on the largest improvement?

Authors' reply: While the improved sampling for V9 is evident year-round, specific months selected for the examples presented in Section 4.2 were chosen because of potential scientific interest in those months. For South America, September is typically the month when CO loading due to biomass burning reaches its peak. For Asia and North America, January also represents a month of relatively high CO loading.  However, other months could have been selected without affecting any of the conclusions of the paper.

p. 15, l. 365: Here I miss a bit of context. What is the reason that the sampling frequency increases sharply over Canada? Do not think aerosols are an issue? So, is this related to albedo? Please provide some additional analysis here.

Authors' reply:  For this manuscript, the specific causes of the increased sampling for V9 over Canada (as indicated in Fig. 6) were not analyzed.  However, a recently published paper by Marey et al. (titled 'Analysis of improvements in MOPITT observational coverage over Canada') analyzed this issue.  The following sentence has been added to the last paragraph of Section 4.2: "Improved sampling for V9 over

Canada was found independently to be related to added retrievals in scenes with low clouds [Marey et al., 2022]."

p. 18, l. 375: Clear, but the TIR product in Figure 4 shows no difference, indicating that MOPITT observed wintertime boundary pollution. Options: (1) make a link in the text to outline this (2) include NIR-TIR analysis in Figure 4 (this option has my preference).

Authors' reply: The area in Fig. 8 showing significant V8/V9 differences is restricted to a 10-by-10 degree area which occupies just a few percent of the area of the corresponding zone (30 N to 40 N) in Fig. 4. Thus, the effect of a strongly localized increase in monthly-mean CO total column of ~ 20% is greatly diminished with respect to its effect on the zonal mean. Figure 4 presents TIR-only data to minimize differences in retrieval sensitivity between ocean and land within each zone (since NIR radiances are only used in daytime/land scenes).